# CausalVTG: Towards Robust Video Temporal Grounding via Causal Inference

**Qiyi Wang**
Tongji University
wqy126179@tongji.edu.cn

**Senda Chen**
Tongji University
2411498@tongji.edu.cn

**Ying Shen**[*]
Tongji University
yingshen@tongji.edu.cn

## Abstract

Video Temporal Grounding (VTG) aims to localize relevant segments in untrimmed videos based on natural language queries and has seen notable progress in recent years. However, most existing methods suffer from two critical limitations. First, they are prone to learning superficial co-occurrence patterns—such as associating specific objects or phrases with certain events—induced by dataset biases, which ultimately degrades their semantic understanding abilities. Second, they typically assume that relevant segments always exist in the video, an assumption misaligned with real-world scenarios where queried content may be absent. Fortunately, causal inference offers a natural solution to the above-mentioned issues by disentangling dataset-induced biases and enabling counterfactual reasoning about query relevance. To this end, we propose CausalVTG, a novel framework that explicitly integrates causal reasoning into VTG. Specifically, we introduce a causality-aware disentangled encoder (CADE) based on front-door adjustment to mitigate confounding biases in visual and textual modalities. To better capture temporal granularity, we design a multi-scale temporal perception module (MSTP) that reconstructs query-conditioned video features at multiple resolutions. Additionally, a counterfactual contrastive learning objective is employed to help the model discern whether a query is truly grounded in a video. Extensive experiments on five widely-used benchmarks demonstrate that CausalVTG outperforms state-of-the-art methods, achieving higher localization precision under stricter IoU thresholds and more accurately identifying whether a query is truly grounded in the video. These results demonstrate both the effectiveness and generalizability of proposed CausalVTG. The code is available at https://github.com/MxLearner/CausalVTG.

## 1 Introduction

With the development of modern Internet and the popularity of video-sharing platforms, videos have surged exponentially, providing a vital medium for information exchange in entertainment, education, and news. This rapid expansion has given rise to challenges in efficient video browsing and retrieval. Video Temporal Grounding (VTG), which aims to localize video segments that semantically correspond to natural language queries, has become a core task in vision-language understanding [1, 2]. It encompasses two representative sub-tasks: *Moment Retrieval* (MR) [3, 4, 5], which focuses on predicting precise temporal boundaries of the target segment, and *Highlight Detection* (HD) [4, 6], which estimates frame-wise saliency scores to identify the most informative and representative video portions. The applications of VTG enhance video content utilization and satisfy the escalating demands for efficient video content acquisition [4, 7, 8].

Moment-DETR is the pioneer work proposed for jointly solving MR and HD tasks along with a unified evaluation benchmark QVHighlights constructed by the authors [4]. Building upon these

---

[*]Corresponding author.

39th Conference on Neural Information Processing Systems (NeurIPS 2025).

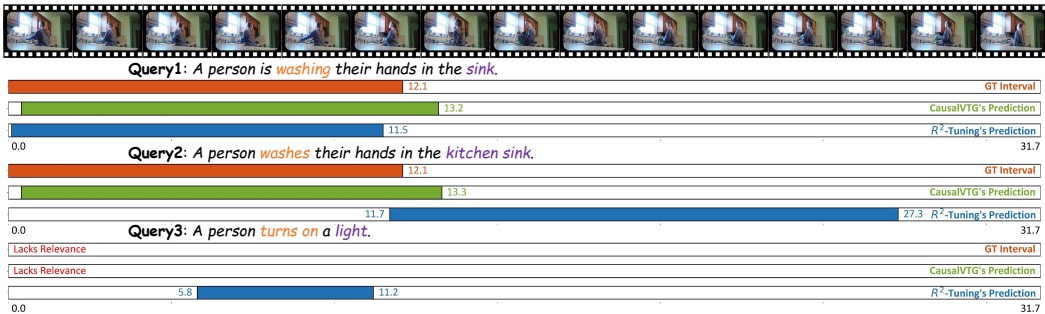

Figure 1: Qualitative examples illustrating two key limitations of existing temporal grounding models. Query1: a successful video grounding example for both VGT models. Query2: a minor rephrasing of the query leads to incorrect localization of $R^2$-Tuning due to reliance on superficial co-occurrence cues (e.g., "kitchen"). Query3: the of $R^2$-Tuning-Tuning model erroneously predicts a segment for a query describing an unseen action, reflecting its inability to assess query relevance. The proposed CausalVTG, better captures temporal causal structures and abstains when appropriate.

efforts, a number of subsequent studies have further advanced the field, notably by leveraging large-scale pre-trained models [4, 5, 9] and incorporating adapter-based techniques [10] to enhance cross-modal alignment. However, the majority of efforts remain centered on improving video-text alignment through increasingly elaborate fusion strategies [11, 12, 13], often overlooking inherent biases embedded in training datasets [4, 14]. These biases include stylistic variations, such as visual decoration patterns and linguistic phrasings [15]. Such factors can lead models to overfit to dataset-specific cues—for instance, correlating background styles or frequent wordings with particular events—rather than learning genuine semantic associations. This spurious reliance ultimately compromises the robustness and generalization ability of temporal grounding systems. For example in Figure 1, given Query1 "A person is washing their hands in the sink," the existing model successfully retrieves the correct moment. Yet, a minor rephrasing as "A person washes their hands in the kitchen sink" leads to incorrect localization of $R^2$-Tuning [10] model on a later segment where the person is drying their hands. This failure illustrates how the model can be misled by superficial visual-textual cues (e.g., "kitchen") instead of reasoning about the actual temporal causal structure of events. Moreover, most existing models operate under the assumption that relevant segments always exist in the video, making them incapable of identifying when a query is actually ungrounded. Given Query3 "A person turns on a light," the $R^2$-Tuning model still returns the above segment based on the presence of a light source, even though the described action does not occur. These limitations reflect a common underlying issue: current approaches are fundamentally correlation-driven and lack the ability to reason about causality, which is essential for robust video temporal grounding.

To address these limitations, we resort to *causal inference* to tackle these challenges. By explicitly modeling the underlying data-generating process, causal inference methods can disentangle spurious correlations and enable counterfactual reasoning—allowing models to focus on genuine semantic causality and assess whether a query is truly grounded in a video. Causal inference [16, 17] has shown promise across machine learning tasks, including image recognition [18, 19], visual question answering [20, 21], and vision-and-language navigation [22, 15]. These methods typically construct a structural causal model [23] to explicitly represent the relationships among variables, and apply causal tools such as the *do-operator*, which simulates interventions to estimate causal effects beyond observational correlations. In addition, techniques like *front-door* and *back-door* adjustments are used to control for latent confounders—either by leveraging mediating variables that transmit causal influence, or by blocking spurious dependencies introduced by common causes. These mechanisms enable the model to better capture true causal relationships, rather than being misled by stylistic artifacts or co-occurrence patterns. Motivated by these successful applications, we investigate how causal principles can be leveraged in VTG, particularly under stylistic variation and uncertain query relevance, two scenarios where conventional methods struggle.

To accomplish these goals, we *firstly* design a **structural causal model** (SCM) specifically tailored to the VTG task to reveal in-depth causal relations, as illustrated in Figure 2. In this model, we explicitly identify stylistic variations, such as visual decoration patterns or linguistic phrasing, as confounding factors that simultaneously influence both visual and textual modalities. To mitigate their

impact, we introduce latent mediators that encode core semantic information from both modalities. These mediators enable the application of front-door adjustment, which reduces the influence of confounding biases by leveraging the causal pathway from inputs to grounding outcomes through the mediators. *Secondly*, building on SCM, we design a **causality-aware disentangled encoder** (CADE) that performs modality-specific front-door adjustments [16] on video and query features. CADE employs mediator-guided attention mechanisms to extract causally disentangled and unbiased representations for each modality, encouraging the model to find out true semantic causalities rather than superficial co-occurrence patterns. *Thirdly*, to further enhance temporal reasoning, we incorporate a **multi-scale temporal perception module** (MSTP), which reconstructs video representations conditioned on the query and projects them across multiple temporal resolutions. This design enables the model to capture both fine-grained and long-range event dynamics, improving localization performance for events of varying durations. *Lastly*, we introduce a **counterfactual contrastive learning objective** [16] to complement the standard VTG training paradigm. This objective enforces discrimination between observed and counterfactual representations at the video level, allowing the model to determine with greater reliability whether a query is semantically grounded in the video. Taken together, these components form a principled and interpretable framework for moment retrieval and highlight detection, offering enhanced robustness against confounding biases and linguistic variations.

Our main contributions are summarized as follows:

- We formulate a structural causal model for VTG that identifies stylistic variations in visual and textual modalities as confounders. By introducing latent mediators and applying front-door adjustment to block spurious dependencies between video/query input and grounding outcomes, our approach establishes a principled causal foundation that improves model robustness and generalization.

- Building on this causal formulation, we design CausalVTG, a unified framework that jointly addresses moment retrieval and highlight detection. Central to our architecture is a causality-aware disentangled encoder which applies front-door adjustment to learn unbiased and semantically meaningful representations for video and query modalities independently. Additionally, we propose a multi-scale temporal perception module to capture hierarchical temporal dynamics, and incorporate a counterfactual contrastive learning objective that enables the model to reliably determine query relevance, improving its ability to reject ungrounded queries.

- We conduct extensive experiments on five widely-used benchmark datasets, demonstrating that CausalVTG consistently outperforms state-of-the-art baselines. Our method achieves significantly higher localization precision under strict IoU thresholds and exhibits superior robustness in discerning whether queries are truly grounded in videos, validating both the effectiveness and generalizability of our causal modeling approach.

## 2  Related Work

**Video Temporal Grounding** VTG focuses on identifying video segments corresponding to natural language queries, encompassing two primary sub-tasks: Moment Retrieval [3, 4, 24, 14, 5] and Highlight Detection [4, 6, 9, 5]. The introduction of the QVHighlights benchmark [4] unified the evaluation of MR and HD tasks and provided a baseline method, Moment-DETR. Subsequent approaches, such as UniVTG [9] and UMT [5], leveraged large-scale vision-language pre-training to enhance model performance. Methods like QD-DETR [11] and CG-DETR [13] emphasized sophisticated cross-modal fusion strategies to improve semantic alignment between video and textual inputs. Approaches including UnLoc [25] and $R^2$-Tuning [10] utilized fine-tuning and transfer learning to adapt CLIP-based models specifically for VTG tasks, whereas LLMEPET [26] integrated large language model encoders into traditional VTG architectures. Recognizing the unrealistic assumption of always-grounded queries, the Charades-RF and ActivityNet-RF datasets [27] introduced scenarios with potentially irrelevant queries. A robust transformer-based framework RaTSG [27] is proposed to explicitly address these false-query situations. However, these models remain fundamentally correlation-driven, vulnerable to confounding biases arising from subtle linguistic and stylistic variations.

**Causal Inference in Video Understanding** Causal inference has emerged as an influential framework for improving generalization in visual understanding by explicitly modeling the underlying data-generating processes and mitigating spurious correlations [16, 23]. Compared to traditional debiasing methods [28, 29], causal approaches leverage counterfactual reasoning to robustly handle domain shifts and biases. In recent computer vision research, causal inference techniques have been applied across tasks such as image classification [18, 19, 30, 31, 32], visual question answering [20, 21, 33], and vision-language navigation [22, 15]. Specifically within the VTG context, DCM [14] utilized back-door adjustments to mitigate temporal biases in video data, while IVG [24] employed both back-door intervention and dual contrastive learning strategies to disentangle misleading visual-textual associations. Beyond VTG, causal inference has also demonstrated effectiveness in video summarization [34] and video captioning [35]. These causal modeling strategies underpin the development of structured causal frameworks for robustly addressing confounding factors such as stylistic variations and irrelevant query relevance in video grounding tasks.

# 3 Preliminary

## 3.1 Task Formulation

Given an input video $\mathcal{V} = \{\mathbf{v}_i\}_{i=1}^{L_v}$ composed of $L_v$ clips, where each clip $\mathbf{v}_i \in \mathbb{R}^{D_v}$ is represented by a $D_v$-dimensional feature vector, and a natural language query $\mathcal{Q} = \{\mathbf{q}_i\}_{i=1}^{L_q}$ consisting of $L_q$ tokens, each with dimension $D_q$, the VTG task comprises three sub-tasks:

**Moment Retrieval** The model predicts temporal segments in the video that semantically align with the query. Since multiple segments may match a single query, the output is a set of predicted moments represented as $\{(b_{s,i}, b_{e,i}, c_i)\}_{i=1}^n$, where $b_{s,i}$ and $b_{e,i}$ denote the start and end timestamps of the $i$-th moment, respectively, and $c_i$ is the associated confidence score for ranking.

**Highlight Detection** The model estimates the semantic relevance between each clip $\mathbf{v}_i$ and the query $\mathcal{Q}$ by assigning a saliency score $s_i \in [0, 1]$. Higher values indicate greater relevance.

**Query Relevance Prediction (QR)** To handle scenarios where the query may not correspond to any content within the video, the model further predicts a query-level relevance score $r \in [0, 1]$. This score quantifies the confidence that the query is grounded in the video, with higher scores indicating higher certainty of relevance, and lower scores reflecting ambiguity or irrelevance.

## 3.2 Structural Causal Model of VTG

As illustrated in Figure 2, we formulate a structural causal model [16] tailored for the VTG task. Specifically, we denote the input video as $\mathcal{V}$, the natural language query as $\mathcal{Q}$, and the grounding outcome as $\mathcal{Y}$, collectively defining inputs as $\mathcal{X} = \{\mathcal{V}, \mathcal{Q}\}$. In this directed acyclic graph, edges represent causal relationships, with input variables $\mathcal{X}$ causing grounding predictions $\mathcal{Y}$. Traditional VTG methods predominantly model observational distributions $P(\mathcal{Y}|\mathcal{X})$, neglecting confounding biases introduced via back-door paths $\mathcal{X} \leftarrow \mathcal{Z} \rightarrow \mathcal{Y}$. Confounders ($\mathcal{Z}$), such as subtle stylistic variations (e.g., visual decoration patterns in videos and linguistic phrasing variations in queries),

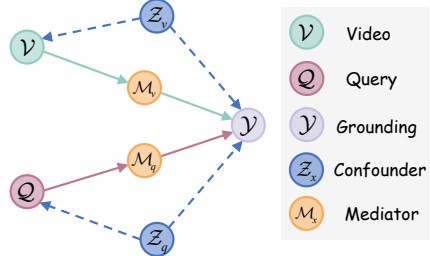

Figure 2: Illustration of the structural causal model of VTG.

simultaneously influence both the input data distribution and grounding outcomes. The presence of these confounding factors inevitably creates spurious correlations, adversely impacting model robustness and generalization.

Since confounders typically comprise intricate and difficult-to-explicitly characterize patterns, we introduce intermediate mediator variables $\mathcal{M}$, establishing explicit front-door causal pathways $\mathcal{X} \rightarrow \mathcal{M} \rightarrow \mathcal{Y}$. This enables effective causal intervention and mitigates the influence of unobserved confounders. The detailed mechanisms for causal adjustment are described in subsequent sections.

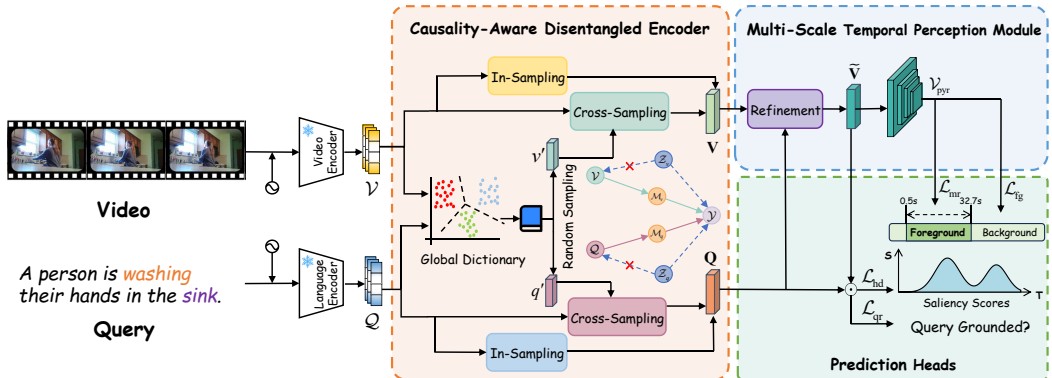

Figure 3: Overview of the CausalVTG framework. Given a video and natural language query, modality-specific encoders extract features and perform front-door adjustment via a causality-aware disentangled encoder to mitigate confounding bias. A query-guided refinement module fuses contextual signals, followed by a multi-scale temporal perception module that captures hierarchical dynamics. Prediction heads jointly perform highlight detection, moment retrieval, and query relevance estimation under a counterfactual training paradigm.

## 4 Methodology

### 4.1 Causality-Aware Disentangled Encoder

Traditional VTG models typically learn the observational distribution $P(\mathcal{Y}|\mathcal{X} = x)$ by directly modeling the correlation between input $\mathcal{X} = \{\mathcal{V}, \mathcal{Q}\}$ and output $\mathcal{Y}$. This process can be expressed as:

$$P(\mathcal{Y}|\mathcal{X} = x) = \sum_{\mathcal{M}=m} P(\mathcal{Y}|\mathcal{M} = m)P(\mathcal{M} = m|\mathcal{X} = x), \tag{1}$$

where $\mathcal{M}$ denotes a set of latent mediators that encode high-level semantic information [36] from $\mathcal{X}$. However, such observational modeling fails to eliminate the influence of confounders $\mathcal{Z}$, which introduce spurious correlations via back-door paths $\mathcal{X} \leftarrow \mathcal{Z} \rightarrow \mathcal{Y}$.

To mitigate this, we adopt a front-door adjustment based on causal intervention using the *do-operator*[16]. Specifically, we introduce a causality-aware disentangled encoder that generates mediators through modality-specific encoders. For each modality (video or text), the encoder computes the mediator representation $m = f(x)$ using a self-attention module [37], which extracts high-level semantic features. Subsequently, the front-door adjustment leverages these mediators to isolate the influence of confounders, enabling the model to focus on genuine causal effects. This constructs a causal path $\mathcal{X} \rightarrow \mathcal{M} \rightarrow \mathcal{Y}$ that permits front-door estimation of the interventional distribution:

$$\begin{aligned} P(\mathcal{Y}|\text{do}(\mathcal{X} = x)) &= \sum_{\mathcal{M}=m} P(\mathcal{Y}|\text{do}(\mathcal{M} = m))P(\mathcal{M} = m|\text{do}(\mathcal{X} = x)) \\ &= \sum_{\mathcal{X}=x'} P(x') \sum_{\mathcal{M}=m} P(\mathcal{M} = m|\mathcal{X} = x)P(\mathcal{Y}|\mathcal{X} = x', \mathcal{M} = m) \\ &= \mathbb{E}_{\mathcal{X}=x'}\mathbb{E}_{\mathcal{M}=m|\mathcal{X}=x}P(\mathcal{Y}|\mathcal{X} = x', \mathcal{M} = m), \end{aligned} \tag{2}$$

where $x'$ denotes inputs sampled from the empirical training distribution via K-means [38], and $m$ is the mediator representation computed from the current input $x$.

To approximate the front-door adjustment, we define a causal feature function modeled as a linear combination of sampled inputs and mediators [15, 39]:

$$\mathcal{F}(x', m) = \mathbb{E}_{\mathcal{X}=x'}\mathbb{E}_{\mathcal{M}=m|\mathcal{X}=x}f(x', m), \quad f(x', m) = h(x') + g(m), \tag{3}$$

which yields the separable form:

$$\mathcal{F}(x', m) = \mathbb{E}_{\mathcal{X}=x'}[h(x')] + \mathbb{E}_{\mathcal{M}=m|\mathcal{X}=x}[g(m)]. \tag{4}$$

We estimate the two expected terms using attention-based mechanisms [40, 14]. Let $x_i'$ and $m_j$ be sampled input and mediator representation from input and mediator distributions, respectively. The expectations are computed as:

$$\mathbb{E}_{\mathcal{X}=x'}[h(x')] = \sum_i \alpha_i h(x_i'), \quad \alpha_i = \frac{\exp(p^\top x_i')}{\sum_k \exp(p^\top x_k')} \tag{5}$$

$$\mathbb{E}_{\mathcal{M}=m|\mathcal{X}=x}[g(m)] = \sum_j \beta_i g(m_j), \quad \beta_j = \frac{\exp(q^\top m_j)}{\sum_k \exp(q^\top m_k)}, \tag{6}$$

where $p = p(x)$ and $q = q(x)$ are attention queries derived from the current input to guide the aggregation of sampled features and mediators.

Through this modality-specific front-door adjustment, we obtain causally disentangled representations for both video and language inputs. These representations are explicitly purified from dataset-induced confounding effects, enabling the model to focus on genuine semantic structures rather than superficial correlations. This serves as a robust foundation for downstream grounding tasks.

## 4.2 Multi-Scale Temporal Perception Module

To enhance the temporal modeling capacity of video features, we design a two-branch perception module comprising: (1) a query-guided refinement and (2) a multi-scale temporal feature pyramid.

**Query-Guided Refinement** Given an input video embedding $\mathbf{V} = [\mathbf{v}_1, \ldots, \mathbf{v}_T] \in \mathbb{R}^{T \times D}$ with $T$ clips and a query embedding $\mathbf{Q} = [\mathbf{q}_1, \ldots, \mathbf{q}_L] \in \mathbb{R}^{L \times D}$ with $L$ tokens, we first compute a similarity matrix $\mathbf{S} \in \mathbb{R}^{T \times L}$ between video and query features using a trilinear attention function [41]:

$$S_{i,j} = \mathbf{v}_i^\top \mathbf{w}_C + \mathbf{q}_j^\top \mathbf{w}_Q + (\mathbf{v}_i \odot \mathbf{w}_M)^\top \mathbf{q}_j, \tag{7}$$

where $\mathbf{w}_C, \mathbf{w}_Q, \mathbf{w}_M \in \mathbb{R}^D$ are learnable parameters and $\odot$ denotes element-wise multiplication.

Based on the similarity matrix $\mathbf{S}$, we compute context-to-query and query-to-context attention as:

$$\mathbf{a}_i^{(Q)} = \sum_{j=1}^{L} \text{softmax}_j(S_{i,j}) \, \mathbf{q}_j, \quad \mathbf{a}_i^{(V)} = \sum_{j=1}^{T} \text{softmax}_i(S_{j,i}) \, \mathbf{v}_j. \tag{8}$$

We then enhance each video clip $\mathbf{v}_i$ by concatenating its interactions with the attended query and video representations:

$$\tilde{\mathbf{v}}_i = \text{Conv1D}\Big( \big[\mathbf{v}_i, \, \mathbf{a}_i^{(Q)}, \, \mathbf{v}_i \odot \mathbf{a}_i^{(Q)}, \, \mathbf{v}_i \odot \mathbf{a}_i^{(V)} \big] \Big), \tag{9}$$

where $\tilde{\mathbf{v}}_i \in \mathbb{R}^D$ is the refined feature of the $i$-th video clip.

This query-guided refinement enables the model to highlight semantically relevant clips and suppress irrelevant noise by embedding the query context directly into the video representation, thereby laying a solid foundation for subsequent temporal reasoning.

**Multi-Scale Temporal Feature Pyramid** To capture hierarchical temporal patterns across different time granularities, we adopt a multi-scale feature pyramid to refine video representations. Given the query-guided video embedding $\tilde{\mathbf{V}} = [\tilde{\mathbf{v}}_1, \ldots, \tilde{\mathbf{v}}_T] \in \mathbb{R}^{T \times D}$, where $T$ is the number of video clips , we construct a set of temporal feature sequences at varying resolutions.

Formally, let $\mathcal{S} = \{s_1, s_2, \ldots, s_K\}$ denote a set of temporal strides, where each $s_k$ represents the downsampling rate for the $k$-th scale. For each scale $s_k \in \mathcal{S}$, we apply a series of temporal convolution layers to obtain:

$$\mathbf{V}^{(k)} = \text{Conv}_{s_k}(\tilde{\mathbf{V}}) \in \mathbb{R}^{T/s_k \times D}, \tag{10}$$

where $\mathbf{V}^{(k)}$ is the video representation at the $k$-th temporal scale.

Each temporal convolution operates along the time dimension and reduces the resolution of the video sequence by a factor of $s_k$. The resulting set of multi-resolution video features $\mathcal{V}_{\text{pyr}} = \{\mathbf{V}^{(1)}, \mathbf{V}^{(2)}, \ldots, \mathbf{V}^{(K)}\}$ forms a hierarchical representation that captures both fine-grained actions and long-term temporal dependencies.

### 4.3 Prediction Heads

We design three task-specific prediction heads to perform highlight detection, moment retrieval, and query relevance prediction. These modules operate on either the refined video features $\tilde{\mathbf{V}}$ or the multi-scale temporal representations $\mathcal{V}_{\text{pyr}}$.

**Highlight Detection** To measure the semantic alignment between video clips and the query, we first apply adaptive average pooling to the query sequence to obtain a global vector: $\mathbf{q}_{\text{ada}} = \text{AdaPool}(\mathbf{Q}) \in \mathbb{R}^D$. Then, for each video clip $\tilde{\mathbf{v}}_i \in \mathbb{R}^D$, we compute its relevance score via cosine similarity:

$$s_i = \frac{\tilde{\mathbf{v}}_i^\top \mathbf{q}_{\text{ada}}}{\|\tilde{\mathbf{v}}_i\| \cdot \|\mathbf{q}_{\text{ada}}\|}, \quad i = 1, \dots, T. \tag{11}$$

The saliency scores $\{s_i\}$ are trained using the SampledNCE loss [42], where positives correspond to informative clips and negatives to unrelated clips.

**Moment Retrieval** We leverage the multi-scale video features $\mathcal{V}_{\text{pyr}} = \{\mathbf{V}^{(1)}, \dots, \mathbf{V}^{(K)}\}$ to localize temporal moments. For each scale $\mathbf{V}^{(k)} \in \mathbb{R}^{T_k \times D}$, two parallel 1D convolutional heads are applied: one for span regression and the other for foreground classification. Specifically, the $\text{Conv1D}_{\text{span}}$ head predicts the relative temporal offsets $\mathbf{b}_i^{(k)} = [d_i^{\text{start}}, d_i^{\text{end}}]$ for each clip $i$, and the $\text{Conv1D}_{\text{fg}}$ head estimates its foreground confidence score $f_i^{(k)} \in \mathbb{R}$:

$$\mathbf{b}_i^{(k)} = \text{Conv1D}_{\text{span}}(\mathbf{V}_i^{(k)}) \in \mathbb{R}^2, \quad f_i^{(k)} = \text{Conv1D}_{\text{fg}}(\mathbf{V}_i^{(k)}) \in \mathbb{R}. \tag{12}$$

The predicted span boundaries $\mathbf{b}_i^{(k)}$ are supervised using L1 loss, while the foreground scores $f_i^{(k)}$ are optimized using focal loss [43] to distinguish query-relevant (foreground) clips from background.

**Query Relevance via Counterfactual Reasoning** To enable query-aware abstention when no grounding evidence exists, we adopt a counterfactual learning strategy. During training, we construct negative samples by pairing a video with a query that is not semantically grounded in it. For both valid and invalid pairs, we compute a joint representation by concatenating the adaptively pooled query and video features:

$$\mathbf{v}_{\text{ada}} = \text{AdaPool}(\tilde{\mathbf{V}}), \quad r = \text{Sigmod}(\text{FFN}([\mathbf{v}_{\text{ada}}; \mathbf{q}_{\text{ada}}])). \tag{13}$$

The predicted score $r \in [0, 1]$ indicates whether the query is relevant to the video, and is optimized using a Dynamic Binary Cross-Entropy loss [44].

### 4.4 Training & Inference

The model is jointly trained with three task-specific objectives. For the HD task, we supervise the clip-wise saliency scores $\{s_i\}$ using the SampledNCE loss [42] $\mathcal{L}_{\text{hd}}$. For the MR task, the predicted boundary offsets $\{\mathbf{b}_i^{(k)}\}$ are optimized with L1 loss $\mathcal{L}_{\text{mr}}$, while the foreground confidence scores $\{f_i^{(k)}\}$ are trained with focal loss [43] $\mathcal{L}_{\text{fg}}$. For the QR task, we use a Dynamic Binary Cross-Entropy loss [44] $\mathcal{L}_{\text{qr}}$ to supervise the scalar prediction $r$ that indicates whether the query is grounded in the video. The overall loss is:

$$\mathcal{L} = \lambda_{\text{hd}}\mathcal{L}_{\text{hd}} + \lambda_{\text{mr}}\mathcal{L}_{\text{mr}} + \lambda_{\text{fg}}\mathcal{L}_{\text{fg}} + \lambda_{\text{qr}}\mathcal{L}_{\text{qr}}. \tag{14}$$

During inference, the saliency scores $\{s_i\}$ directly indicate clip-level highlight relevance. For MR, each predicted span $\mathbf{b}_i^{(k)}$ is converted into a temporal interval $(b_{s,i}, b_{e,i})$ with confidence score $c_i = \sigma(f_i^{(k)})$; we perform Non-Maximum Suppression (NMS) [45] to obtain final ranked results. The query relevance score $r \in [0, 1]$ determines whether to output any grounding result, enabling the model to abstain in irrelevant cases.

## 5 Experiments

### 5.1 Datasets & Evaluation Metrics

Experiments are conducted on five benchmarks: QVHighlights [4] annotated for both MR and HD, serving as a comprehensive benchmark for multi-task evaluation; Charades-STA [3] and ActivityNet-

Table 1: Performance comparison on the QVHighlights test set for joint MR and HD tasks.

| Method | MR | | | | | HD | |
|---|---|---|---|---|---|---|---|
| | R1@0.5 | R1@0.7 | mAP Avg. | mAP@0.5 | mAP@0.75 | mAP | HIT@1 |
| MCN [48] | 11.41 | 2.72 | 10.67 | 24.94 | 8.22 | - | - |
| CAL [49] | 25.49 | 11.54 | 9.89 | 23.40 | 7.65 | - | - |
| XML [50] | 41.83 | 30.35 | 32.14 | 44.63 | 31.73 | 34.49 | 55.25 |
| XML+ [4] | 46.69 | 33.46 | 34.90 | 47.89 | 34.67 | 35.38 | 55.06 |
| Moment-DETR [4] | 52.89 | 33.02 | 30.73 | 54.82 | 29.40 | 35.69 | 55.60 |
| UMT [5] | 56.23 | 41.18 | 36.12 | 53.83 | 37.01 | 38.18 | 59.99 |
| MomentDiff [51] | 58.21 | 41.48 | 36.84 | 54.57 | 37.21 | - | - |
| QD-DETR [11] | 62.40 | 44.98 | 39.86 | 62.52 | 39.88 | 38.94 | 62.40 |
| MH-DETR [12] | 60.05 | 42.48 | 38.38 | 60.75 | 38.13 | 38.22 | 60.51 |
| UniVTG [9] | 58.86 | 40.86 | 35.47 | 57.60 | 35.59 | 38.20 | 60.96 |
| TR-DETR [52] | 64.66 | 48.96 | 42.62 | 63.98 | 43.73 | 39.91 | 63.42 |
| CG-DETR [13] | 65.43 | 48.38 | 42.86 | 64.51 | 45.77 | 40.33 | **66.21** |
| $R^2$-Tuning [10] | 68.03 | 49.35 | 46.17 | 69.04 | 47.56 | **40.75** | 64.20 |
| **CausalVTG (Ours)** | **68.87** | **52.53** | **49.63** | **70.70** | **51.77** | 40.66 | 65.63 |

Caption [46] annotated with precise temporal segments for MR; and Charades-RF and ActivityNet-RF [27] extend their original datasets by introducing false-query scenarios for evaluating whether the query is grounded.

For evaluation metrics, we follow previous work [10, 9]. On QVHighlights, we report performance on both MR and HD. For the MR task, we use mean Average Precision (mAP) at Intersection-over-Union (IoU) thresholds $\theta_{IoU} \in \{0.5, 0.75\}$, mAP averaged over thresholds from 0.5 to 0.95 (with a step size of 0.05), and Recall@1 at $\theta_{IoU} \in \{0.3, 0.5, 0.7\}$. For the HD task, we adopt mAP and HIT@1, considering a clip as positive if it is labeled as "Very Good". On Charades-STA and ActivityNet-Caption, we evaluate MR performance using Recall@1 at $\theta_{IoU} \in \{0.3, 0.5, 0.7\}$ and mean IoU (mIoU). On Charades-RF and ActivityNet-RF, we follow the dataset splits defined in [27] and use Recall@1 at $\theta_{IoU} \in \{0.3, 0.5, 0.7\}$, mIoU, and grounding accuracy (Acc) to measure both localization quality and the model's ability to identify whether a query is truly grounded in the video.

## 5.2 Implementation Details

All models are implemented using PyTorch and trained for 50 epochs on a single NVIDIA RTX 4070 SUPER GPU (12GB VRAM, 32GB RAM). InternVideo2-CLIP [47] serves as the unified backbone for both video and text encoding, with all hidden dimensions set to 256. The temporal stride set for the multi-scale feature pyramid is $\mathcal{S} = \{1, 2, 4, 8\}$ for standard-length videos, and extended to $\{1, 2, 4, 8, 16\}$ for longer videos in ActivityNet-Caption and ActivityNet-RF. The training objective is a weighted combination of four losses, with coefficients $\lambda_{hd} = 0.1$, $\lambda_{mr} = 0.2$, $\lambda_{fg} = 1.0$, and $\lambda_{qr} = 0.1$. Training on the QVHighlights dataset takes approximately 50 minutes.

## 5.3 Comparison with State-of-the-Arts

**Joint Moment Retrieval and Highlight Detection on QVHighlights** We evaluate our model on the QVHighlights test set for the joint tasks of MR and HD, with results summarized in Table 1. CausalVTG achieves state-of-the-art performance across all MR metrics, including notable improvements in R1@0.7 and mAP@0.75, indicating stronger robustness under strict temporal alignment. We attribute these gains to our causal modeling design, which mitigates the influence of spurious correlations. Our method also performs competitively on HD, demonstrating the effectiveness of unified, causally grounded temporal grounding.

**Moment Retrieval on Charades-STA and ActivityNet-Caption** We conduct moment retrieval experiments on Charades-STA and ActivityNet-Caption, with results shown in Table 2. CausalVTG achieves the best performance across all metrics on both datasets. Notably, on Charades-STA, where each video segment is annotated with multiple queries exhibiting diverse linguistic expressions—our model demonstrates substantial improvements. We attribute this to the causal modeling framework,

Table 2: Moment retrieval results on Charades-STA and ActivityNet-Caption test sets.

| Method | Charades-STA | | | | ActivityNet Captions | | | |
|---|---|---|---|---|---|---|---|---|
| | R1@0.3 | R1@0.5 | R1@0.7 | mIoU | R1@0.3 | R1@0.5 | R1@0.7 | mIoU |
| VSLNet [53] | 67.47 | 54.62 | 35.43 | 49.37 | 62.12 | 43.76 | 25.64 | 44.54 |
| SeqPAN [54] | 70.70 | 59.14 | 41.02 | 52.32 | _63.71_ | _45.31_ | **26.69** | _45.73_ |
| TCN+DCM [14] | - | 55.8 | 34.4 | 48.7 | - | 44.9 | 27.7 | 43.3 |
| DORi [55] | 72.72 | 59.65 | 40.56 | 53.28 | 57.89 | 41.35 | _26.41_ | 42.79 |
| EAMAT [56] | _74.25_ | _61.18_ | _41.72_ | _54.53_ | 62.20 | 41.60 | 24.14 | 44.15 |
| ADPN [57] | 71.24 | 56.88 | 39.73 | 51.96 | 61.46 | 41.49 | 24.78 | 44.12 |
| QD-DETR [11] | 70.32 | 58.92 | 38.54 | 50.62 | 62.20 | 41.60 | 24.14 | 44.15 |
| UniVTG [9] | 71.62 | 60.06 | 33.34 | 49.92 | 61.78 | 43.34 | 22.59 | 42.71 |
| $R^2$-Tuning [10] | 70.91 | 59.78 | 37.02 | 50.86 | - | - | - | - |
| RaTSG [27] | 74.19 | 56.61 | 37.47 | 53.02 | 61.46 | 42.36 | 24.74 | 43.72 |
| **CausalVTG (Ours)** | **81.37** | **70.89** | **49.25** | **59.96** | **64.44** | **45.62** | 26.28 | **45.74** |

Table 3: Performance comparison on Charades-RF and ActivityNet-RF datasets.

| Method | Charades-RF | | | | | ActivityNet-RF | | | | |
|---|---|---|---|---|---|---|---|---|---|---|
| | Acc | R1@0.3 | R1@0.5 | R1@0.7 | mIoU | Acc | R1@0.3 | R1@0.5 | R1@0.7 | mIoU |
| VSLNet [53] | 50.00 | 33.74 | 27.31 | 17.72 | 24.69 | 50.00 | 31.06 | 21.88 | 12.82 | 22.27 |
| UniVTG [9] | 50.00 | 35.81 | 30.03 | 16.67 | 24.96 | 50.00 | 30.89 | 21.67 | 11.29 | 21.35 |
| QD-DETR [11] | 50.00 | 35.16 | 29.46 | 19.27 | 25.31 | 50.00 | 26.50 | 19.15 | 11.07 | 18.99 |
| ADPN [57] | 50.00 | 35.62 | 28.44 | 19.87 | 25.98 | 50.00 | 30.72 | 20.74 | 12.38 | 22.05 |
| SeqPAN [54] | 50.00 | 35.35 | 29.57 | 20.51 | 26.14 | 50.00 | 31.85 | 22.65 | 13.34 | 22.86 |
| EAMAT [56] | 50.00 | 37.12 | 30.59 | 20.86 | 27.27 | 50.00 | 31.10 | 20.80 | 12.07 | 22.07 |
| VSLNet[++] | 71.94 | 61.40 | 56.77 | 49.65 | 54.67 | 81.60 | 66.15 | 58.37 | 50.64 | 58.65 |
| UniVTG[++] | 71.94 | 62.58 | 58.55 | 48.79 | 54.65 | 81.60 | 66.15 | 58.36 | 49.46 | 58.00 |
| QD-DETR[++] | 71.94 | 62.18 | 58.20 | 50.96 | 55.13 | 81.60 | 62.43 | 56.13 | 49.27 | 55.97 |
| ADPN[++] | 71.94 | 62.26 | 57.23 | 51.16 | 55.41 | 81.60 | 65.85 | 57.41 | 50.28 | 58.47 |
| SeqPAN[++] | 71.94 | 62.12 | 58.01 | 51.61 | 55.49 | 81.60 | 66.77 | 58.98 | 51.11 | 59.11 |
| EAMAT[++] | 71.94 | 63.55 | 59.17 | 51.96 | 56.23 | 81.60 | 66.13 | 57.36 | 49.93 | 58.45 |
| RaTSG [27] | _76.85_ | _68.17_ | _61.91_ | _54.19_ | _59.93_ | _84.27_ | _69.02_ | _60.68_ | _52.88_ | _61.15_ |
| **CausalVTG (Ours)** | **84.78** | **76.22** | **71.07** | **61.03** | **67.86** | **89.20** | **72.70** | **63.62** | **54.64** | **63.82** |

which effectively mitigates the impact of stylistic variations and enhances temporal grounding accuracy.

**Moment Retrieval with Query Relevance on Charades-RF and ActivityNet-RF** We evaluate our model on Charades-RF and ActivityNet-RF, which introduce ungrounded queries to test model robustness. As prior methods assume all queries are grounded, they yield a chance-level accuracy of 50% under the balanced (1:1) positive-negative split. For fair comparison, we implement the ++ versions of baselines by equipping them with a trivial relevance discriminator following previous studies [27]. RaTSG represents the current state-of-the-art. As shown in Table 3, CausalVTG outperforms all baselines by a large margin across both accuracy and localization metrics. These improvements highlight the effectiveness of our causal modeling in identifying true grounding conditions and rejecting spurious matches.

## 5.4 Ablation Study

To thoroughly evaluate the contribution of each component in CausalVTG, we conduct a comprehensive ablation study on the QVHighlights validation set. The analysis decomposes CausalVTG into its four core modules: the Causality-Aware Disentangled Encoder (CADE), Query-Guided Refinement (QGR), Multi-Scale Temporal Perception (MSTP), and Query Relevance Module (QRM). Starting from a baseline model without these modules, we progressively integrate each component individually and jointly. As shown in Table 4, CADE and MSTP yield the most significant gains, demonstrating the effectiveness of causal disentanglement and temporal multi-scale modeling. The addition of

QGR further enhances video-language alignment, while QRM improves the model's ability to handle irrelevant queries. These results validate the complementary benefits of each proposed module.

Table 4: Comprehensive ablation study on the QVHighlights validation set. Each configuration shows the incremental integration of CADE (Causality-Aware Disentangled Encoder), QGR (Query-Guided Refinement), MSTP (Multi-Scale Temporal Perception), and QRM (Query Relevance Module).

| Model Variant | Modules | | | | Performance Metrics | | | | |
|:---:|:---:|:---:|:---:|:---:|:---:|:---:|:---:|:---:|:---:|
| | CADE | QGR | MSTP | QRM | R1@0.5 | R1@0.7 | mAP@0.5 | mAP@0.75 | Avg. mAP |
| (a) | | | | | 57.74 | 36.52 | 58.96 | 35.36 | 35.19 |
| (b) | ✓ | | | | 59.94 | 39.55 | 60.22 | 37.43 | 36.47 |
| (c) | | ✓ | | | 60.39 | 38.52 | 60.39 | 37.05 | 36.61 |
| (d) | | | ✓ | | 67.68 | 51.68 | 69.48 | 51.33 | 47.85 |
| (e) | | | | ✓ | 60.77 | 39.16 | 61.32 | 36.98 | 36.87 |
| (f) | ✓ | ✓ | | | 61.32 | 38.58 | 61.65 | 37.22 | 37.11 |
| (g) | ✓ | | ✓ | | 68.58 | 52.71 | 69.69 | 50.89 | 48.99 |
| (h) | ✓ | | | ✓ | 62.19 | 40.05 | 61.66 | 37.86 | 37.42 |
| (i) | ✓ | ✓ | ✓ | | 68.13 | 52.90 | 69.95 | 52.00 | 49.54 |
| (j) | ✓ | | ✓ | ✓ | 70.26 | 54.32 | 71.34 | 52.67 | 50.15 |
| (k) | ✓ | ✓ | ✓ | ✓ | **70.84** | **56.00** | **72.17** | **53.79** | **50.98** |

# 6   Conclusion

This paper introduces CausalVTG, a causal framework for Video Temporal Grounding that addresses the limitations of existing correlation-driven methods. By modeling structural causal relationships among video, query, and grounding outcomes, the framework effectively mitigates spurious correlations caused by dataset biases. Key components include a causality-aware disentangled encoder for front-door adjustment, a multi-scale temporal perception module for capturing hierarchical dynamics, and a counterfactual contrastive learning objective for reliable query relevance estimation. Experimental results across five benchmarks confirm the effectiveness and generalizability of the proposed approach in both standard and challenging grounding scenarios.

## Acknowledgments

This work was supported in part by the National Natural Science Foundation of China under Grant 62476202 and 62272343, in part by Shanghai Pujiang Program (No. 23PJ1412700), in part by the Fundamental Research Funds for the Central Universities.

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

# A Technical Appendices and Supplementary Material

## A.1 Details of the Datasets

**QVHighlights** [4] is the only dataset supporting both moment retrieval and highlight detection. It contains 10,148 trimmed videos spanning diverse domains such as daily vlogs, travel, and news. A total of 10,310 queries are annotated with 18,367 disjoint temporal moments. It also includes a private test split for benchmarking.

**Charades-STA** [3] augments the Charades dataset with temporal annotations. It consists of 9,848 indoor videos averaging 30.6 seconds in duration, annotated with 16,128 query-moment pairs.

**ActivityNet-Captions** [46] is built upon ActivityNet v1.3 and contains over 20,000 untrimmed videos with 100,000 sentence-level annotations. Each video averages 120 seconds in length and includes multiple annotated events. The dataset is split into 10,024 training, 4,926 validation, and 5,044 test videos.

**Charades-RF** and **ActivityNet-RF** [27] are extensions of Charades-STA and ActivityNet-Captions that introduce ungrounded (false) queries to evaluate query relevance prediction. Each query is either grounded or intentionally mismatched to test the model's abstention capability.

## A.2 Sensitivity to the Number of K-means Clusters and Clustering Randomness

To investigate the robustness of the proposed CADE framework with respect to the number of K-means clusters and clustering randomness, we conduct a systematic ablation on the Charades-RF [27] dataset by varying the number of clusters $K \in \{16, 32, 64, 128, 256, 512, 1024, 2048\}$. For each $K$, multiple random seeds are used to initialize the K-means algorithm, and we report the mean $\pm$ standard deviation of Recall@1 at IoU = 0.7 (R1@0.7), mean IoU (mIoU), and grounding accuracy (Acc).

Table 5: Ablation on the number of K-means clusters $K$ on Charades-RF. Mean $\pm$ std over multiple seeds.

| Metric / $K$ | 16 | 32 | 64 | 128 | 256 | 512 | 1024 | 2048 |
|---|---|---|---|---|---|---|---|---|
| R1@0.7 | $58.93 \pm 1.71$ | $60.39 \pm 0.66$ | $\mathbf{61.25 \pm 0.31}$ | $60.29 \pm 0.40$ | $58.20 \pm 0.41$ | $59.12 \pm 0.89$ | $59.25 \pm 0.53$ | $59.06 \pm 0.45$ |
| mIoU | $65.58 \pm 1.30$ | $66.18 \pm 0.68$ | $\mathbf{67.16 \pm 0.07}$ | $66.44 \pm 0.37$ | $64.36 \pm 0.54$ | $65.64 \pm 0.33$ | $65.56 \pm 0.16$ | $65.15 \pm 0.43$ |
| Acc | $84.25 \pm 0.57$ | $85.11 \pm 0.14$ | $\mathbf{85.76 \pm 0.69}$ | $84.96 \pm 0.51$ | $83.50 \pm 0.15$ | $84.37 \pm 0.28$ | $84.00 \pm 0.51$ | $84.11 \pm 0.11$ |

As shown in Table 5, the model achieves the best overall performance when $K = 64$, where all three metrics peak with low standard deviation, indicating stable behavior across different random seeds. Using too few clusters ($K \leq 32$) leads to under-segmentation of the latent semantic space, resulting in insufficient modeling of confounders and a notable drop in grounding performance. In contrast, using too many clusters ($K \geq 512$) introduces noisy and overly fine-grained mediator representations, increasing computational cost and reducing performance stability. The standard deviation remains relatively small for $K$ values in the range of $[32, 128]$, which demonstrates that CADE maintains robustness to K-means initialization randomness within this moderate range. This analysis confirms that a balanced cluster granularity (e.g., $K = 64$) provides the best trade-off between semantic coverage and stability.

## A.3 Sensitivity to Temporal Strides in MSTP

We evaluate the effect of multi-scale temporal proposals (MSTP) on the QVHighlights validation set by varying the temporal stride set used to generate proposals. We compare a model without MSTP to configurations using strides $\{1\}$, $\{1, 2\}$, $\{1, 2, 4\}$, $\{1, 2, 4, 8\}$, and $\{1, 2, 4, 8, 16\}$. As summarized in Table 6, introducing multi-scale temporal modeling substantially improves recall and average precision; performance increases monotonically from single- to four-scale settings, indicating that capturing a range of action durations is critical for precise event localization, with an additional coarse scale (16) providing complementary gains.

Table 6: Sensitivity analysis of temporal strides used in MSTP on the QVHighlights validation set.

| Temporal strides | R1@0.5 | R1@0.7 | mAP@0.5 | mAP@0.75 | Avg.mAP |
|---|---|---|---|---|---|
| w/o MSTP | 61.35 | 38.00 | 61.40 | 36.75 | 36.29 |
| {1} | 60.84 | 40.58 | 62.15 | 37.81 | 37.56 |
| {1,2} | 67.10 | 48.45 | 67.69 | 45.16 | 43.35 |
| {1,2,4} | 70.39 | 53.10 | 70.91 | 50.00 | 48.63 |
| {1,2,4,8} | **70.84** | **56.00** | 72.17 | **53.79** | 50.98 |
| {1,2,4,8,16} | 70.19 | 54.97 | **72.33** | 53.76 | **51.88** |

## A.4 Computational Cost and Efficiency

We evaluate computational overhead on the QVHighlights dataset using an NVIDIA A800 GPU (80GB) with batch size 64 for 50 epochs. As summarized in Table 7, CausalVTG's inference runtime is slightly longer than simpler baselines yet remains competitive. Notably, prior methods such as Moment-DETR and QD-DETR originally required up to 200 epochs, and $R^2$-Tuning utilized extensive GPU memory in training due to the reversed recurrent block.

Table 7: Computational cost comparison on QVHighlights.

| Method | GPU Memory | #Parameters | Training Time | Inference Time |
|---|---|---|---|---|
| Moment-DETR [4] | 1.41 GB | 4.82 M | 9.71 min | 31 s |
| QD-DETR[11] | 1.89 GB | 7.58 M | 13.15 min | 37 s |
| CG-DETR [13] | 3.09 GB | 12.61 M | 40.05 min | 45 s |
| $R^2$-Tuning [10] | 37.24 GB | 2.7 M | 544.33 min | 69 s |
| CausalVTG | 2.31 GB | 7.86 M | 43.21 min | 53 s |

## A.5 More Visualizations

To further assess model behavior, qualitative comparisons are conducted on the QVHighlights [4] validation set against the strong baseline $R^2$-Tuning [10]. As illustrated in Figure 4, CausalVTG yields more precise and consistent results in both moment retrieval and highlight detection across various scenarios, reflecting better temporal alignment and semantic understanding. To analyze failure cases, Figure 5 presents examples where both $R^2$-Tuning and CausalVTG do not produce accurate predictions. These often involve fine-grained visual attributes such as subtle differences in clothing color or object type, which remain challenging due to limited perceptual cues.

## A.6 Limitations & Future Work

While CausalVTG demonstrates strong performance in temporal grounding, it currently leverages only visual and textual modalities. In certain scenarios, especially those involving speech-centric queries or audio-specific events, incorporating audio cues could provide critical complementary information. Moreover, the model still struggles with fine-grained distinctions such as subtle visual attributes or small object interactions, as shown in the failure cases. Future work could explore multimodal extensions with audio and depth signals, as well as finer perceptual modeling to improve grounding in visually ambiguous or low-resolution settings.

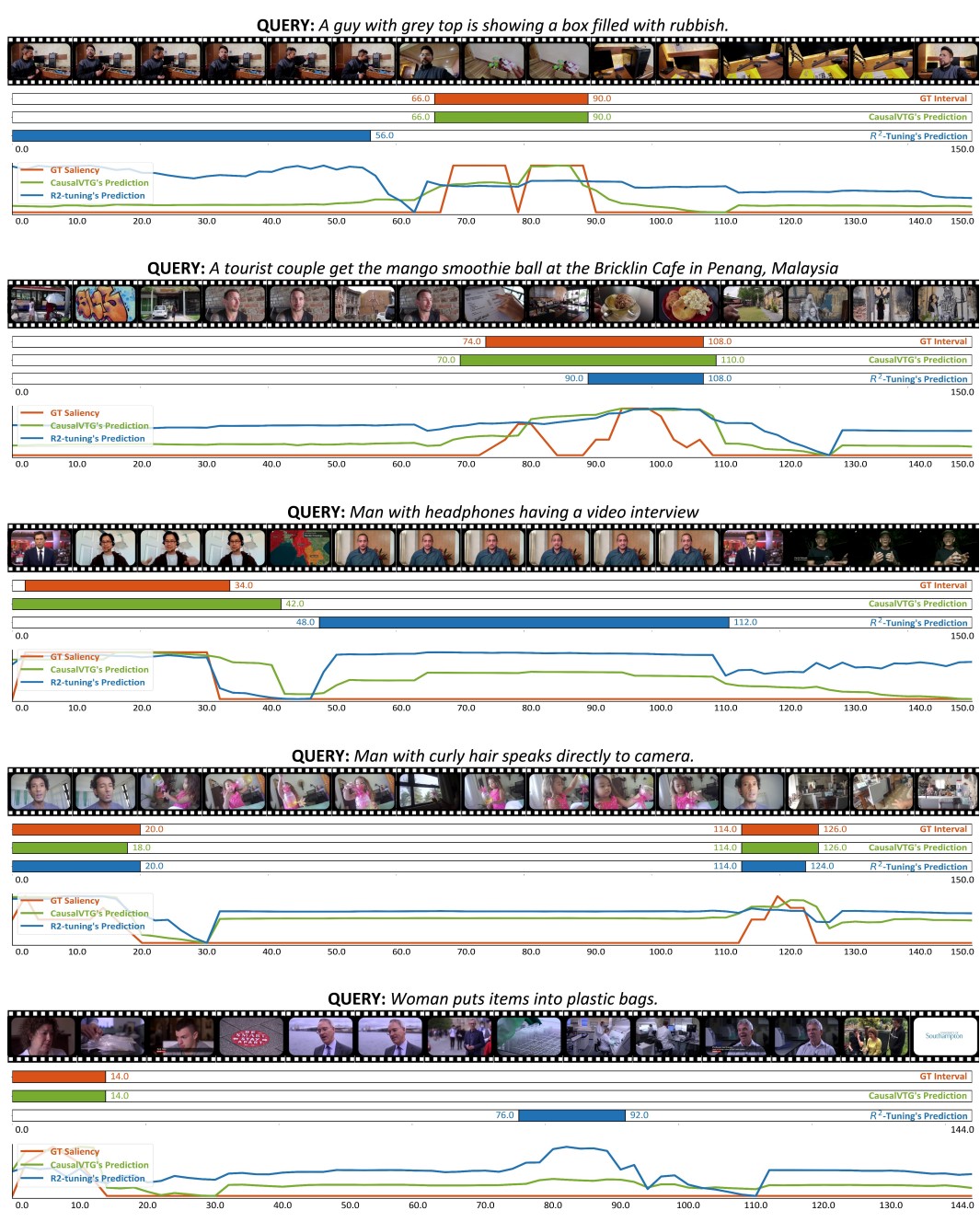

Figure 4: Qualitative results showing successful predictions by CausalVTG compared to $R^2$-Tuning. CausalVTG more accurately captures the target moments and saliency.

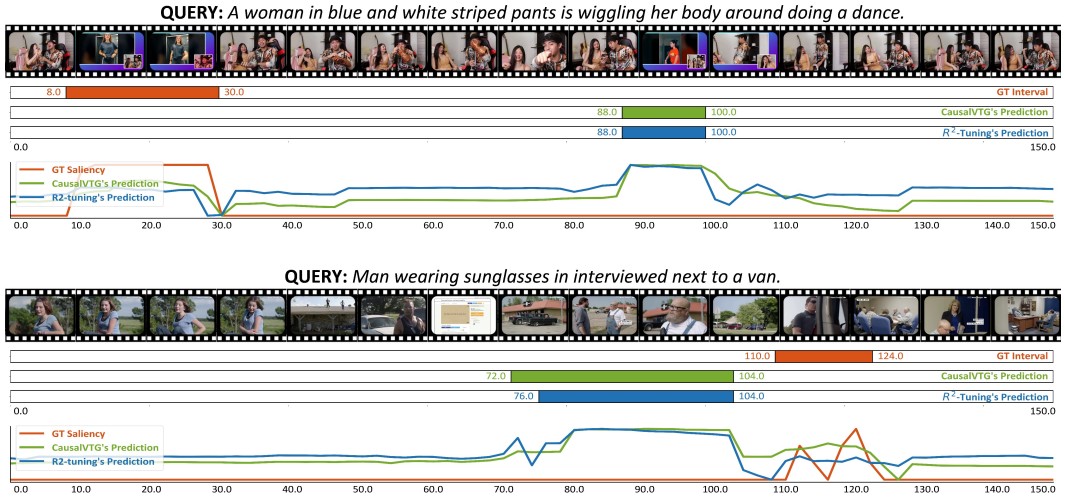

Figure 5: Failure cases where CausalVTG struggles with fine-grained visual distinctions, such as clothing color or object details.

