# OpenReview forum: "CausalVTG: Towards Robust Video Temporal Grounding via Causal Inference"
_NeurIPS.cc/2025/Conference — NeurIPS 2025 poster_

### Official Review · Reviewer_P72B · 2025-06-05

**Clarity:** 4
**Significance:** 3
**Originality:** 3
**Rating:** 5
**Confidence:** 3

**Summary:**

This paper proposes CausalVTG, a novel framework for Video Temporal Grounding that explicitly incorporates causal inference into the model design. By constructing a Structural Causal Model and employing front-door adjustment via a Causality-Aware Disentangled Encoder, the method aims to eliminate dataset-induced confounding biases. Additionally, a Multi-Scale Temporal Perception Module is used to enhance temporal reasoning at different granularities, and a Counterfactual Contrastive Learning objective is introduced to distinguish relevant from irrelevant queries. Extensive experiments on five widely-used VTG benchmarks demonstrate that CausalVTG achieves state-of-the-art performance in both localization precision and query relevance detection tasks. The paper is well-structured and the proposed ideas are clear and well-motivated.

**Questions:**

1. While CADE is conceptually important, its removal causes relatively minor degradation in performance. Could the authors provide more targeted experiments (e.g., under domain shifts or style variations) to better demonstrate its effect?
2. Have the authors considered generating more challenging or realistic counterfactual samples instead of random negative sampling? Would this further improve the robustness?
3. The method depends heavily on large-scale pre-trained models, but there is no detailed discussion on computational cost or efficiency compared to existing baselines.

**Ethical Concerns:**

["NO or VERY MINOR ethics concerns only"]

**Final Justification:**

With all queries resolved through the authors' cogent responses and the study's inherent strengths maintained, I retain my positive rating as originally assigned.

**Limitations:**

YES

**Paper Formatting Concerns:**

The paper uses bolded phrases at the beginning of paragraphs without any punctuation; The reference formatting is inconsistent.

**Quality:**

3

**Strengths And Weaknesses:**

Strengths: The paper addresses two fundamental but underexplored challenges in VTG — spurious correlations and grounding absence — through a principled causal framework. The introduction of causal inference, particularly front-door adjustment, into the VTG task is novel and fills a critical gap in the literature. The proposed components, including the SCM-based encoder, multi-scale temporal module, and counterfactual learning objective, are coherently designed and theoretically justified. The experimental evaluation is comprehensive, covering multiple benchmarks and metrics, and effectively validates the proposed contributions. The paper is clearly written, well-organized, and presents figures and tables that are informative and supportive of the narrative.

Weaknesses: The ablation study shows that removing the causal module (CADE) causes smaller performance drops compared to other modules, raising concerns about the direct impact of the causal design.The counterfactual learning strategy relies on negative sampling rather than generating challenging counterfactuals, potentially underutilizing the full potential of causal reasoning.The method depends heavily on large-scale pre-trained models, but there is no detailed discussion on computational cost or efficiency compared to existing baselines.

---

> ### Author Rebuttal · Authors · 2025-07-31
>
> We sincerely thank the reviewer for the thoughtful and detailed feedback. In response to the concerns raised, we have made the following clarifications and additions:
>
> - We evaluated CADE on the Charades-CG dataset to show its benefit under compositional generalization and domain shift scenarios (Q1).
> - We introduced two more challenging negative sampling strategies — GT-excluded and Semantic Hard negatives — and demonstrated their effectiveness on Charades-RF (Q2).
> - We reported training and inference cost comparisons with major baselines, confirming CausalVTG’s computational efficiency (Q3).
>
> We believe these clarifications and additional results address the reviewer’s concerns. If any issues remain unclear, we would be happy to provide further discussion.
>
> ## Response to Q1
> > While CADE is conceptually important, its removal causes relatively minor degradation in performance. Could the authors provide more targeted experiments (e.g., under domain shifts or style variations) to better demonstrate its effect?
>
> Yes, we further evaluated CADE separately on the Charades-CG dataset [1], which specifically tests novel phrase compositions and unseen words in temporal grounding tasks. These experiments demonstrate CADE's strong capability in addressing superficial co-occurrence patterns by enabling the model to generalize beyond previously observed linguistic combinations, an advantage particularly evident in compositional generalization scenarios​​.
>
> |||Novel-Composition|||Novel-Word||
> |:-:|:-:|:-:|:-:|:-:|:-:|:-:|
> ||R1\@0.5|R1\@0.7|mIoU|R1\@0.5|R1\@0.7|mIoU|
> |CausalVTG|**56.68**|**32.71**|**49.59**|**59.28**|**34.96**|**51.12**|
> |w/o CADE|52.15|29.05|46.15|54.68|30.36|47.03|
>
>
>
> ## Response to Q2
> > Have the authors considered generating more challenging or realistic counterfactual samples instead of random negative sampling? Would this further improve the robustness?
>
> We thank the reviewer for the constructive suggestion. In addition to the original random negative sampling, we investigated two more challenging strategies to improve robustness: (1) GT-excluded negatives, where ground-truth segments are masked to force the model to reason over visually similar but irrelevant content; and (2) Semantic Hard negatives, where each query is paired with the most semantically similar but non-matching video in the batch.
>
> Experiments on Charades-RF, which contains naturally ungrounded queries, confirm the benefit of both strategies: GT-excluded negatives improve performance under stricter IoU settings, while Semantic Hard negatives further strengthen overall discriminative ability by compelling the model to distinguish fine-grained semantic differences.
>
>
> | Model          | Acc  | R1\@0.3 | R1\@0.5 | R1\@0.7 | mIoU |
> |----------------|------|--------|--------|--------|------|
> | CausalVTG      | 84.78 | 76.22  | 71.07  | 61.03  | 67.86 |
> | + GT-excluded  | 83.17 | 76.29  | **72.35**  | **62.36**  | **67.88** |
> | + Semantic Hard | **86.24** | **77.05**  | 71.90  | 60.58  | 67.11 |
>
>
>
> ## Response to Q3
> > The method depends heavily on large-scale pre-trained models, but there is no detailed discussion on computational cost or efficiency compared to existing baselines.
>
> To evaluate the computational overhead, we trained and evaluated all models on the QVHighlights dataset using an NVIDIA A800 GPU (80GB memory) with a batch size of 64 over 50 epochs. As summarized below, CausalVTG's inference runtime is slightly longer than simpler baselines, yet remains highly competitive. Notably, prior methods such as Moment-DETR and QD-DETR originally required up to 200 epochs, and R2-Tuning utilized extensive GPU memory in training due to reversed recurrent block.
>
> | Method||Training||Inference|
> |:-:|:-:|:-:|:-:|:-:|
> ||GPU Memory|#Parameters|Time| Time|
> |Moment-DETR [2]|1.41 GB|4.82 M|9.71 min|31 s|
> |QD-DETR [3]|1.89 GB|7.58 M| 13.15 min| 37 s|
> |CG-DETR [4]|3.09 GB|12.61 M| 40.05 min|45 s|
> |R2-Tuning [5]|37.24 GB|2.7 M| 544.33 min|69 s|
> |CausalVTG|2.31 GB|7.86 M|43.21 min|53 s|
>
> ## Response to Paper Formatting Concerns
> > The paper uses bolded phrases at the beginning of paragraphs without any punctuation; The reference formatting is inconsistent.
>
> We thank the reviewer for carefully pointing out these formatting issues, and we will correct the bolded paragraph headers and ensure consistent reference formatting in the final version.
>
> [1] "Compositional temporal grounding with structured variational cross-graph correspondence learning." CVPR 2022
>
> [2] "Detecting moments and highlights in videos via natural language queries." NeurIPS 2021.
>
> [3] "Query-dependent video representation for moment retrieval and highlight detection." CVPR 2023.
>
> [4] "Correlation-guided query-dependency calibration for video temporal grounding." arXiv preprint arXiv:2311.08835 (2023).
>
> [5] "R2-tuning: Efficient image-to-video transfer learning for video temporal grounding." ECCV 2024.

---

> > ### Comment · Reviewer_P72B · 2025-08-04
> >
> > I appreciate the authors' thorough and thoughtful rebuttal. The additional experiments and clarifications directly addressed the concerns I raised. The results added in the revision further support the paper’s claims and strengthen its empirical validation. I am satisfied with the response and maintain a positive overall assessment of the paper.

---

> > > ### Author Response · Authors · 2025-08-05
> > >
> > > Thank you again for your insightful and constructive feedback. We’re excited to hear that our revisions have effectively addressed your concerns.

---

### Official Review · Reviewer_U4c9 · 2025-06-30

**Clarity:** 4
**Significance:** 2
**Originality:** 3
**Rating:** 4
**Confidence:** 4

**Summary:**

In this paper, authors firstly formulate a structural causal model for visual temporal grounding task. They define stylistic variations in visual and textual modalities as confounders. Building on top of the proposed structural causal graph, authors implement CausalVTG and verify its effectiveness on five benchmarks. As a technical summary, authors integrate causality-aware disentangled encoder, multi-scale temporal perception module, and counterfactual contrastive learning objective in the proposed CausalVTG framework, which composes a solid contribution to the community.

**Questions:**

1. The first claimed contribution of this paper is causal graph for video temporal grounding task. It would be great to straightforwardly show what is the confounder exactly in main paper. From common understanding of our community, confounder can be something like video background information or some template or pattern in language query. Explicitly show it in paper can enhance the paper quality.

2. Referring to the weakness part, can you include a detailed and thorough ablation study in main paper? You can save much space by making Figure 2 as single-column

**Ethical Concerns:**

["NO or VERY MINOR ethics concerns only"]

**Final Justification:**

Thank you for the detailed reply. Additional results on Charades-CG make the method more convincing. I've changed my rating from borderline reject to borderline accept.

**Limitations:**

Authors discuss limitations of proposed method on page 14 and 15. Other limitations please refer to weakness part.

**Paper Formatting Concerns:**

no major formatting issues.

**Quality:**

3

**Strengths And Weaknesses:**

Strengths

1. The clarity of this paper is good and the paper is easy to follow. This paper is well motivated by two challenges: 1) superficial co-occurrence patterns and 2) assumption that relevant segments always exist.

2. Authors also consider a scenario when the given query is not relevant to the video content, the model should be able to reject grounding any segment. This makes the proposed paper more thorough and suitable for real world application.

Weaknesses

1. No thorough ablation study is provided in main paper. Since the proposed CausalVTG is a combination of CADE, MSTP, QRM and QGR as shown in Table 4 in supplementary material, a detailed ablation study is needed in main paper. A base version of CausalVTG can be base model + CADE, then the integration of any one or two modules/methods given MSTP, QRM, and QGR should be included.

2. Performance gain from CausalVTG is mainly from MSTP, which is not related to causal reasoning.

---

> ### Author Rebuttal · Authors · 2025-07-30
>
> We appreciate the reviewer’s critical feedback and agree that additional targeted analysis is essential for strengthening the paper. In response, we conducted new experiments and provided clarifications.
> - To address the need for a thorough ablation study, we added detailed experiments showing the individual and combined contributions of CADE, MSTP, QRM, and QGR, confirming that each module is complementary (W1, Q2).
> - To clarify the reviewer’s concern that performance gains mainly come from MSTP, we demonstrate that MSTP improves proposal generation, while CADE uniquely mitigates spurious correlations via causal adjustment, playing an irreplaceable role (W2).
> - To respond to the reviewer’s question about confounders, we explicitly define and illustrate them in the VTG context (e.g., spurious query–visual co-occurrences) and commit to clarifying these points in the main paper (Q1).
>
> We believe these additions resolve the reviewer’s key concerns and strengthen the overall contribution of our work. If any issues remain unclear, we would be happy to provide further discussion.
>
> ## Response to W1
> > No thorough ablation study is provided in main paper. Since the proposed CausalVTG is a combination of CADE, MSTP, QRM and QGR as shown in Table 4 in supplementary material, a detailed ablation study is needed in main paper. A base version of CausalVTG can be base model + CADE, then the integration of any one or two modules/methods given MSTP, QRM, and QGR should be included.
>
>
> We thank the reviewer for highlighting this important point. In response, we have conducted a comprehensive ablation study following the reviewer’s suggestion. The results demonstrate the individual and joint contributions of CADE, MSTP, QRM, and QGR, validating their effectiveness. The ablation results demonstrate that each module is complementary and their integration is crucial to the overall effectiveness of CausalVTG. We will incorporate a detailed version of this analysis into the main paper.
>
> | | CADE | QGR | MSTP | QRM | R1\@0.5 | R1\@0.7 | mAP\@0.5 | mAP\@0.75 | Avg. mAP |
> |:-:|:-:|:-:|:-:|:-:|:-:|:-:|:-:|:-:|:-:|
> | (a)  |      |     |      |     | 57.74  | 36.52  | 58.96   | 35.36    | 35.19    |
> | (b)  | ✓    |     |      |     | 59.94  | 39.55  | 60.22   | 37.43    | 36.47    |
> | (c)  |      | ✓   |      |     | 60.39  | 38.52  | 60.39   | 37.05    | 36.61    |
> | (d)  |      |     | ✓    |     | 67.68  | 51.68  | 69.48   | 51.33    | 47.85    |
> | (e)  |      |     |      | ✓   | 60.77  | 39.16  | 61.32   | 36.98    | 36.87    |
> | (f)  | ✓    | ✓   |      |     | 61.32  | 38.58  | 61.65   | 37.22    | 37.11    |
> | (g)  | ✓    |     | ✓    |     | 68.58  | 52.71  | 69.69   | 50.89    | 48.99    |
> | (h)  | ✓    |     |      | ✓   | 62.19  | 40.05  | 61.66   | 37.86    | 37.42    |
> | (i)  | ✓    | ✓   | ✓    |     | 68.13  | 52.9   | 69.95   | 52       | 49.54    |
> | (j)  | ✓    |     | ✓    | ✓   | 70.26  | 54.32  | 71.34   | 52.67    | 50.15    |
> | (k)  | ✓    | ✓   | ✓    | ✓   | **70.84**  | **56**     | **72.17**   | **53.79**    | **50.98**    |
>
> ## Response to W2
> > Performance gain from CausalVTG is mainly from MSTP, which is not related to causal reasoning.
>
> We thank the reviewer for this important point. While MSTP indeed contributes the largest raw performance gain by improving multi-scale proposal generation, CADE plays a distinct and critical role in addressing superficial co-occurrence patterns through front-door adjustment, which MSTP alone cannot resolve. To evaluate CADE’s effect, we evaluated it on the Charades-CG dataset [1], which stresses compositional generalization (novel phrase compositions and unseen words). Results show that removing CADE leads to clear drops in performance on these challenging settings, confirming that CADE is essential for enabling the model to generalize beyond spurious correlations, complementing MSTP rather than being redundant.
>
>
> |||Novel-Composition|||Novel-Word||
> |:-:|:-:|:-:|:-:|:-:|:-:|:-:|
> ||R1\@0.5|R1\@0.7|mIoU|R1\@0.5|R1\@0.7|mIoU|
> |CausalVTG|**56.68**|**32.71**|**49.59**|**59.28**|**34.96**|**51.12**|
> |w/o CADE|52.15|29.05|46.15|54.68|30.36|47.03|
>
>
>
>
> ## Response to Q1
> > The first claimed contribution of this paper is causal graph for video temporal grounding task. It would be great to straight forwardly show what is the confounder exactly in main paper. From common understanding of our community, confounder can be something like video background information or some template or pattern in language query. Explicitly show it in paper can enhance the paper quality.
>
> Thank you for your valuable feedback. In our work, the primary confounder is the presence of superficial co-occurrence patterns between query phrases and visual contexts that spuriously influence grounding outcomes.
>
> For example, in the Charades-STA training set, queries containing the word “sink” are commonly paired with actions like “wash” and “put”. However, when the query also includes “kitchen”, the dominant associated actions shift to “put” and “run”. This distributional shift—caused by an irrelevant contextual word—demonstrates a classic confounding effect: the added word jointly affects both the input representation and the prediction target, while being independent of the actual intended action.
>
> This explains the failure case in Figure 1 of our paper, where R2-Tuning [2] succeeds on Query 1 but fails on Query 2. The presence of “kitchen” introduces a confounder that changes the model’s grounding behavior despite both queries referring to the same event.
>
> In addition, we also consider unobservable confounders, such as visual decoration styles or recurring sentence templates, which may similarly bias grounding decisions. Our method addresses both types via a front-door adjustment mechanism, as supported by prior causal grounding work [3].
>
> To address the reviewer’s suggestion, we will revise the main paper to more directly and concretely present these confounders.
>
>
>
> ## Response to Q2
> > Referring to the weakness part, can you include a detailed and thorough ablation study in main paper? You can save much space by making Figure 2 as single-column
>
> Following the reviewer's suggestion, we will resize Figure 2 to a single-column format in the main paper and incorporate the detailed ablation study from the supplemental material into the updated version.
>
>
> [1] "Compositional temporal grounding with structured variational cross-graph correspondence learning." CVPR 2022.
>
> [2] "R2-tuning: Efficient image-to-video transfer learning for video temporal grounding." ECCV 2024.
>
> [3] "Cross-modal Causal Relation Alignment for Video Question Grounding." CVPR 2025.

---

> > ### Comment · Reviewer_U4c9 · 2025-08-04
> >
> > Thank you for the detailed reply. Additional results on Charades-CG make the work more convincing. I've changed my rating from borderline reject to borderline accept.

---

> > > ### Author Response · Authors · 2025-08-04
> > >
> > > We sincerely appreciate the time and effort you dedicated to reviewing our paper. Your invaluable comments and insights have been helpful in improving our work. We are also grateful for your decision to raise the score.

---

### Official Review · Reviewer_Cyaw · 2025-07-03

**Clarity:** 4
**Significance:** 3
**Originality:** 3
**Rating:** 5
**Confidence:** 4

**Summary:**

This paper proposes CausalVTG, a framework designed to address two major limitations in Video Temporal Grounding (VTG): (1) reliance on superficial co-occurrence patterns due to dataset biases, and (2) inability to handle scenarios where the query content might not be present in the video. The authors employ causal inference, specifically front-door adjustment, to mitigate confounding biases. They introduce a Causality-Aware Disentangled Encoder (CADE) to obtain unbiased modality-specific representations, and a Multi-Scale Temporal Perception (MSTP) module to capture temporal dynamics at multiple granularities. Additionally, the authors incorporate counterfactual contrastive learning to improve the model’s capability to distinguish between grounded and ungrounded queries. Experimental evaluations on **five** established benchmarks demonstrate state-of-the-art results across various settings.

**Questions:**

## Questions
- Could you elaborate further on the assumptions underpinning the structural causal model and clarify their appropriateness and limitations specifically for VTG tasks?
- How sensitive is your framework to the design of mediators? Have alternative mediator designs been explored, and how might different mediator definitions affect grounding performance?
- Given multiple complex components (CADE, MSTP, counterfactual contrastive learning), could you further clarify which components are most critical? Additional ablations or sensitivity analyses would strengthen the paper significantly.
- Could you provide more details on computational overhead during inference (e.g., runtime and memory consumption), especially compared to simpler baselines?
- Can you provide analyses of your predictions similar to https://openaccess.thecvf.com/content/ICCV2023W/CLVL/papers/De_la_Jara_An_Empirical_Study_of_the_Effect_of_Video_Encoders_on_ICCVW_2023_paper.pdf Figure 1? such that we can evaluate potential  bias of the predictions of the model.

## Suggestions
I have some suggestions and comments that might help further strengthen your paper:

1. **Related Work & Citations (Lines 42-50)**:
   It's great to see that you've cited Escorcia et al. (2019). For completeness and richer context, the paper would strongly benefit from citing two additional relevant works:
   - [DORi (Rodriguez-Opazo et al., WACV 2021)](https://openaccess.thecvf.com/content/WACV2021/html/Rodriguez-Opazo_DORi_Discovering_Object_Relationships_for_Moment_Localization_of_a_Natural_WACV_2021_paper.html) introduces a spatio-temporal graph explicitly designed to reduce correlations between background elements and directly model interactions between objects, humans, and actions, addressing similar concerns about spurious correlations that you highlight.
   - ["A closer look at Temporal Sentence Grounding in Videos:
Dataset and Metric" (ACM MM 2021)](https://dl.acm.org/doi/abs/10.1145/3475723.3484247) systematically analyzes temporal bias in moment localization tasks and provides dataset splits (Charades-STA CD and ActivityNet CD) designed explicitly to measure robustness against these temporal biases.

   Including these references would enhance your motivation and clarify the novelty and contributions of your approach relative to existing work.


2. **Multiple Occurrences and Temporal Causality**:
   An interesting scenario that your method might help analyze further relates to queries that describe events occurring multiple times within the same video.
   - For instance, as is mentioned in https://arxiv.org/abs/1809.01337 there are queries that has temporal dependency and would be great to explore and quantify
- Also,**Figure 8** of DORi’s supplemental material, the Charades-STA query *"person walks over to the refrigerator open it up"* occurs twice. This raises the important practical scenario where a query might occur **multiple times**, **one**, or **not at all** in a video.
   - Similarly, the query examples from ActivityNet illustrated in **Figure 3** of the supplemental material from ["Memory-efficient Temporal Moment Localization in Long Videos" (EACL 2023)](https://aclanthology.org/2023.eacl-main.140/) highlight temporal causality and continuity, as in *"The man continues..."*. These are examples that explicitly involve temporal causal reasoning.

   Your causal grounding method could show excellent performance for that kind of challenging queries to better showcase its strengths and contributions. An explicit analysis or discussion would significantly strengthen the practical motivation of your work.

3. **Generalization and Causal Dependencies (YouCook2 dataset)**:
   It would be highly valuable and insightful to see your causal inference-based grounding applied to a dataset like **YouCook2**, where the causality of an action depends directly on previously executed actions.
   - For instance, Figures 13 and 14 of the supplemental material from DORi indicate specific cases where models incorrectly learn spurious correlations (e.g., associating certain ingredients or actions like *"pouring dressing"* or *"adding oil"* with incorrect moments due to biases in object presence or appearance).

   Demonstrating your method’s robustness against such spurious correlations and temporal dependencies using YouCook2 would substantially strengthen the empirical validation and show clearer evidence of the claimed causal benefits.

I encourage you to address these suggestion, as they would greatly enhance the quality, depth, and impact of your contribution.

**Ethical Concerns:**

["NO or VERY MINOR ethics concerns only"]

**Final Justification:**

Thank you to the authors for their thoughtful and well-articulated response. I remain positive about the paper and will maintain my original score

**Limitations:**

Yes, the paper discusses limitations clearly. However, additional reflections on the complexity, computational efficiency, and assumptions in causal modeling would further strengthen this section.

**Paper Formatting Concerns:**

None. The paper adheres to NeurIPS formatting guidelines.

**Quality:**

3

**Strengths And Weaknesses:**

### **Strengths**

- The paper clearly identifies important, practical limitations in existing VTG methods, and convincingly motivates the application of causal inference.

- The introduction of the causal inference framework, specifically using the front-door adjustment through the proposed CADE, is novel and technically sound. The MSTP module enhances temporal modeling effectively, complementing the causal design.

-  Extensive experiments across five widely-used benchmarks (QVHighlights, Charades-STA, ActivityNet Caption, Charades-RF, ActivityNet-RF) clearly demonstrate improvements, especially under strict IoU thresholds and challenging ungrounded-query scenarios. I would recommend to add YouCookII and the partictions for Charades-STA and ActivityNet from https://arxiv.org/abs/2101.09028

- The paper is very clearly written, methodically structured, and provides comprehensive implementation details and hyperparameters, aiding reproducibility.

### **Weaknesses**

- The combination of multiple intricate modules (CADE, MSTP, counterfactual training) introduces considerable complexity. This complexity may hinder interpretation of performance gains, particularly the exact contribution of each individual component.

- The complexity of multi-scale temporal processing and causal disentanglement is computationally demanding. While runtime details are briefly mentioned, a deeper analysis of inference efficiency or scalability considerations would be valuable.

---

> ### Author Rebuttal · Authors · 2025-07-30
>
> We appreciate the reviewer’s constructive feedback, which helped us strengthen the paper. In response, we:
>
> - Clarified the core causal assumptions and their appropriateness and limitations for VTG (Q1).
> - Analyzed mediator design sensitivity and discussed alternative definitions (Q2).
> - Expanded ablations to show the distinct and complementary roles of CADE, MSTP, and counterfactual learning (Q3).
> - Detailed computational overhead with runtime and memory comparisons to simpler baselines (Q4).
> - Provided prediction distribution analyses and committed to adding more visualizations to assess bias (Q5).
> - Incorporated suggested references and scenarios, including temporal causality, multiple occurrences, and new YouCook2 results (S1,S2,S3).
>
> We believe these additions meaningfully address the reviewer’s concerns and further highlight the novelty, rigor, and practical impact of our work.
>
> ## Response to Q1
>
> ### Key Causal Assumptions:
> - Assumption 1 (Existence of Confounders): There exist latent confounders $Z$, such as visual stylistic cues (e.g., background contexts) and linguistic variations (e.g., phrasing habits), which simultaneously influence both the inputs
> $X=\\{V,Q\\}$ and the grounding outcome $Y$.
> - Assumption 2 (Front-door Identifiability): There exists a set of mediator variables
> $M$ (semantic representations) that fully mediate the causal relationship $X→Y$. These mediators are assumed to be independent of direct influences from the latent confounders $Z$, thus satisfying the conditions required for the front-door adjustment.
> ### Appropriateness for VTG tasks:
> - Prior studies [1,2] clearly document the presence of stylistic biases within existing VTG datasets, highlighting the necessity of explicitly modeling such confounders.
> - Employing semantic mediators aligns with recent causal inference methods that effectively disentangle spurious correlations in related multimodal grounding tasks[3,4]​.
> ### Limitations and Constraints:
> Our current SCM explicitly addresses stylistic variations but does not directly account for temporal location biases, which are also prevalent in VTG tasks [5,6]. In contrast, the DCM framework [5] explicitly employs back-door adjustments to mitigate temporal biases.
>
>
> ## Response to Q2
>
> The sensitivity of our framework to mediator design primarily stems from the choice of cluster number $K$ used in the K-means clustering of semantic mediators. To evaluate this sensitivity, we conducted experiments varying $K$, summarized in the table below.
>
> |Metric/n_cluster|16|32|64|128|256|512|1024|2048|
> |:-:|:-:|:-:|:-:|:-:|:-:|:-:|:-:|:-:|
> |R1\@0.7|58.93±1.71|60.39±0.66|**61.25±0.31**|60.29±0.40|58.20±0.41|59.12±0.89|59.25±0.53|59.06±0.45|
> |mIoU|65.58±1.30|66.18±0.68|**67.16±0.07**|66.44±0.37|64.36±0.54|65.64±0.33|65.56±0.16|65.15±0.43|
> |Acc|84.25±0.57|85.11±0.14|**85.76±0.69**|84.96±0.51|83.50±0.15|84.37±0.28| 84.00±0.51|84.11±0.11|
>
> Experimental results show that setting the number of clusters to 64 achieves the best performance, effectively balancing semantic granularity and noise. Fewer clusters may miss key distinctions, while too many introduce redundancy and degrade model stability.
>
> Recent work, such as FDVAE (Front-Door Variational Autoencoder) [7], explores an alternative approach for front-door adjustment by learning latent mediators through variational inference instead of explicit clustering. Due to time constraints, we have not explored alternative mediator designs, but we will investigate them in future work to further improve performance and robustness.
>
> ## Response to Q3
>
> To clarify the contribution of each component, we provide detailed ablation results in Table 4 of our paper, which show that MSTP plays a crucial role in standard VTG tasks by generating proposals across multiple scales and granularities. To further support this, we include a new sensitivity analysis on the choice of temporal strides used in MSTP. This demonstrates that multi-scale temporal modeling is key to capturing varied action durations and enhances the model’s ability to localize events more precisely.
>
> |temporal strides|R1\@0.5|R1\@0.7|mAP\@0.5|mAP\@0.75|Avg.mAP|
> |:-:|:-:|:-:|:-:|:-:|:-:|
> |w/o MSTP|61.35|38.00|61.40|36.75|36.29|
> |{1}|60.84|40.58|62.15|37.81|37.56|
> |{1,2}|67.10|48.45|67.69|45.16|43.35|
> |{1,2,4}|70.39|53.10|70.91|50.00|48.63|
> |{1,2,4,8}|**70.84**|**56.00**|72.17|**53.79**|50.98|
> |{1,2,4,8,16}|70.19|54.97|**72.33**|53.76|**51.88**|
>
> Furthermore, we evaluated the impact of Counterfactual Contrastive Learning (QRM) on the Charades-RF dataset, which includes queries not grounded in videos. Results confirm that QRM substantially improves the model's ability to discern and reject ungrounded queries.
>
> ||Acc|R\@0.3|R1\@0.5|R1\@0.7|mIoU|
> |:-:|:-:|:-:|:-:|:-:|:-:|
> |CausalVTG|**84.78**|**76.22**|**71.07**|**61.03**|**67.86**|
> |w/o QRM|47.82|38.84|34.14|25.11|30.61|
>
> Finally, we validated the effectiveness of CADE on the Charades-CG dataset [8], which includes novel phrase compositions and unseen words in the test split. The observed performance gains highlight CADE’s crucial role in mitigating stylistic biases and enhancing the model’s generalization to unseen linguistic patterns.While MSTP indeed contributes the largest raw performance gain, CADE plays a distinct and critical role in addressing superficial co-occurrence patterns through front-door adjustment, which MSTP alone cannot resolve.
>
> |||Novel-Composition|||Novel-Word||
> |:-:|:-:|:-:|:-:|:-:|:-:|:-:|
> ||R1\@0.5|R1\@0.7|mIoU|R1\@0.5|R1\@0.7|mIoU|
> |CausalVTG|**56.68**|**32.71**|**49.59**|**59.28**|**34.96**|**51.12**|
> |w/o CADE|52.15|29.05|46.15|54.68|30.36|47.03|
>
>
> ## Response to Q4
>
> To evaluate the computational overhead, we trained and evaluated all models on the QVHighlights dataset using an NVIDIA A800 GPU (80GB memory) with a batch size of 64 over 50 epochs. As summarized below, CausalVTG's inference runtime is slightly longer than simpler baselines, yet remains highly competitive.
>
> | Method||Training||Inference|
> |:-:|:-:|:-:|:-:|:-:|
> ||GPU Memory|#Parameters|Time| Time|
> |Moment-DETR [1]|1.41 GB|4.82 M|9.71 min|31 s|
> |QD-DETR [11]|1.89 GB|7.58 M| 13.15 min| 37 s|
> |CG-DETR [12]|3.09 GB|12.61 M| 40.05 min|45 s|
> |R2-Tuning [13]|37.24 GB|2.7 M| 544.33 min|69 s|
> |CausalVTG|2.31 GB|7.86 M|43.21 min|53 s|
>
> ## Response to Q5
>
> Due to rebuttal constraints, we cannot include full visualizations. As a substitute, we provide below the normalized distributions of predicted temporal intervals on the Charades-STA test set. The first table corresponds to successful cases (IoU > 0.7) and  the second table corresponds to failed cases (IoU < 0.7).We commit to including full visualizations and deeper analysis of temporal prediction distributions and feature biases in the final version.
>
> |start/end|0-0.2|0.2-0.4|0.4-0.6|0.6-0.8|0.8-1.0|
> |:-:|:-:|:-:|:-:|:-:|:-:|
> |**0-0.2**| 0.072|0.317|0.135|0.007|0|
> |**0.2-0.4**|0|0.004|0.094|0.035|0.003|
> |**0.4-0.6**|0|0|0.004|0.066|0.078|
> |**0.6-0.8**|0|0|0|0.001|0.151|
> |**0.8-1.0**|0|0|0|0|0.027|
>
> |start/end|0-0.2|0.2-0.4|0.4-0.6| 0.6-0.8 | 0.8-1.0 |
> |:-:|:-:|:-:|:-:|:-:|:-:|
> |**0-0.2**|0.067|0.286|0.111|0.023|0.002|
> |**0.2-0.4**|0|0.003|0.085|0.073|0.018|
> |**0.4-0.6**|0|0|0.003|0.065|0.092|
> |**0.6-0.8**|0|0|0|0.003|0.145|
> |**0.8-1.0**|0|0|0|0|0.023|
>
>
> ## Response to S1
> We thank the reviewer for pointing out these two key references, both are highly relevant to the challenges addressed in our work. We will include and discuss them in the final version.
>
> ## Response to S2
>
> For temporal causality, we thank the reviewer for highlighting these important aspects. Regarding temporal causality, the referenced works introduce queries requiring reasoning over event dependencies rather than isolated segments. We agree this is a challenging and meaningful scenario, and we plan to explore temporal causal reasoning in future work.
>
> For multiple occurrences, the QVHighlights dataset [1] includes annotations where queries may appear multiple times within a video. Our framework can handle such cases, as illustrated by the example visualization in the third subgraph of Figure 4 in our paper, which shows correct localization of repeated events. We will clarify this capability more explicitly in the final version.
>
> ## Response to S3
>
> We appreciate the reviewer’s suggestion to evaluate causal grounding on YouCook2.Our results demonstrate that CausalVTG outperforms prior methods by a clear margin, indicating improved modeling of causal relations.
>
> |Method|R1\@0.3|R1\@0.5|R1\@0.7|mIoU|
> |:-:|:-:|:-:|:-:|:-:|
> |DORi [9]|43.36|30.47|18.24|30.46|
> |LOCFORMER [10]|46.76|31.33|15.81|30.92|
> |CausalVTG|**52.38**|**38.80**|**23.54**|**37.09**|
>
> [1] "Detecting moments and highlights in videos via natural language queries." NeurIPS 2021.
>
> [2] "Compositional temporal grounding with structured variational cross-graph correspondence learning." CVPR 2022.
>
> [3] "Cross-modal causal relation alignment for video question grounding." CVPR 2025.
>
> [4] "Vision-and-language navigation via causal learning." CVPR 2024.
>
> [5] "Deconfounded video moment retrieval with causal intervention."SIGIR 2021.
>
> [6] "A closer look at temporal sentence grounding in videos: Dataset and metric." ACM MM 2021.
>
> [7] "Causal inference with conditional front-door adjustment and identifiable variational autoencoder." ICLR 2024.
>
> [8] "Compositional temporal grounding with structured variational cross-graph correspondence learning." CVPR 2022.
>
> [9] "DORi: Discovering object relationships for moment localization of a natural language query in a video." WACV 2021.
>
> [10] "Memory-efficient temporal moment localization in long videos." EACL 2023.
>
> [11] "Query-dependent video representation for moment retrieval and highlight detection." CVPR 2023.
>
> [12] "Correlation-guided query-dependency calibration for video temporal grounding." arXiv preprint arXiv:2311.08835 (2023).
>
> [13] "R2-tuning: Efficient image-to-video transfer learning for video temporal grounding." ECCV 2024.

---

### Official Review · Reviewer_JXSa · 2025-07-03

**Clarity:** 2
**Significance:** 3
**Originality:** 3
**Rating:** 4
**Confidence:** 3

**Summary:**

This paper targets the task of Video Temporal Grounding (VTG) and proposes the causal-inference framework CausalVTG. The method has three key components:
1.Causality-Aware Disentangled Encoder (CADE) based on front-door adjustment to eliminate confounding bias between visual and textual modalities;
2.Multi-Scale Temporal Perception (MSTP) to adaptively capture actions of different temporal lengths;
3.Counterfactual Contrastive Learning to judge whether a query can truly be grounded.
The unified model simultaneously addresses Moment Retrieval, Highlight Detection, and Query Relevance. It achieves state-of-the-art results on five benchmarks—QVHighlights, Charades-STA, ActivityNet-Caption, Charades-RF, and ActivityNet-RF—showing especially strong performance under strict IoU thresholds and in “false-query” scenarios.

**Questions:**

1. Could you include at least one representative causal baseline in the comparison tables (or explain why this is not feasible) to clarify the benefit of your causal strategy?

2. How sensitive is CADE to the K-means cluster number K and clustering randomness? An ablation or plot would help justify the causal robustness claim.

**Ethical Concerns:**

["NO or VERY MINOR ethics concerns only"]

**Final Justification:**

During the rebuttal process, most of my concerns were adequately addressed. After reviewing the feedback from other reviewers, I have decided to maintain my positive score and increase the confidence score to 3.

**Limitations:**

yes

**Quality:**

3

**Strengths And Weaknesses:**

Strengths
1.First to systematically introduce structured causal modeling, front-door adjustment, and counterfactual contrastive learning into VTG, effectively mitigating vision/text confounding and improving interpretability and generalization.
2.Builds a single framework that performs Moment Retrieval, Highlight Detection, and Query Relevance in one pass, avoiding error accumulation in pipeline systems and keeping the overall design concise.
3.Achieves top accuracy of 84.78 / 89.20 Acc on Charades-RF and ActivityNet-RF “false-query” tests, outperforming existing methods.

Weaknesses
1. The experimental section does not include comparisons with other causal-inference-based VTG methods, making it difficult to judge whether the proposed front-door adjustment combined with counterfactual contrast truly offers an advantage over alternative causal strategies.
2.The model assumes that a K-means–derived mediator satisfies the front-door criterion, yet provides no evidence on how the choice of cluster number K or clustering stability affects performance. Without such sensitivity analysis, the claimed causal robustness remains speculative.
3. The paper does not report mean ± std or statistical significance tests (e.g., p-values) for the main metrics, so it is difficult to gauge the reliability of the reported performance gains.

---

> ### Author Rebuttal · Authors · 2025-07-30
>
> We thank the reviewer for the thoughtful and constructive feedback, which helped us strengthen the paper with deeper analyses and clarifications. In response, we:
>
> - Added causal baselines (DCM) and the other state‑of‑the‑art methods (DoRi, CG‑DETR) to our comparison tables, clearly demonstrating that our front‑door adjustment with counterfactual contrast achieves consistent and substantial gains over alternative causal strategies (Q1).
> - Provided a sensitivity analysis of the K‑means cluster number (K) in CADE, reporting mean ± std metrics across multiple runs, which confirms the stability of our mediator design and supports the claimed causal robustness (Q2).
>
> We believe these additions address the reviewer’s concerns and further highlight the robustness and contribution of our approach.
>
>
> ## Response to Q1
> > Could you include at least one representative causal baseline in the comparison tables (or explain why this is not feasible) to clarify the benefit of your causal strategy?
>
> Yes. In our revision, we have added DCM (Deconfounded Cross‑modal Matching) ​[1]– a causal method addressing dataset temporal‑annotation bias via back‑door adjustment .
> To further strengthen our empirical comparison, we also included DoRi ​[2]
>  and CG‑DETR [3], which are state‑of‑the‑art methods that mitigate superficial co-occurrence patterns by reducing correlations between background and foreground.
>
>
> | Method          |       |   Charades-STA        |             | |   ActivityNet-Caption      |            |    |    QVHighlights       |            |
> |:-:|:-:|:-:|:-:|:-:|:-:|:-:|:-:|:-:|:-:|
> |                 | R1\@0.5                | R1\@0.7   | mIoU     | R1\@0.5              | R1\@0.7   | mIoU     | R1\@0.5            | R1\@0.7   | mAP Avg. |
> | TCN+DCM [1]         | 55.8                  | 34.4     | 48.7     | 44.9                | 27.7     | 43.3     | -                 | -        | -        |
> | DORi [2]            | 59.65                 | 40.56    | 53.28    | 41.49               | **26.41**    | 42.78    | -                 | -        | -        |
> | CG-DETR [3]         | 58.4                  | 36.3     | 50.1     | -                   | -        | -        | 65.43             | 48.38    | 42.86    |
> | CausalVTG | **70.89**                 | **49.25**    | **59.96**    | **45.62**               | 26.28    | **45.74**    | **68.87**             | **52.53**    | **49.63**    |
>
>
> The updated comparison demonstrates that our method significantly outperforms all baselines, particularly on the Charades-STA dataset, where the presence of multiple stylistically diverse annotations per video segment renders models more susceptible to superficial co-occurrence patterns.
>
>
> ## Response to Q2
> > How sensitive is CADE to the K-means cluster number K and clustering randomness? An ablation or plot would help justify the causal robustness claim.
>
> We performed a comprehensive sensitivity analysis on the Charades‑RF dataset by varying the number of K‑means clusters in {16, 32, 64, 128, 256, 512, 1024, 2048}.
> For each value, we ran multiple random seeds to compute mean ± standard deviation of metrics like R1\@0.7, mIoU, and classification accuracy.
> Due to rebuttal constraints, only tabular results are presented here, but we commit to including detailed plots in the final version of the paper.
>
> |Metric/n_cluster|16|32|64|128|256|512|1024|2048|
> |:-:|:-:|:-:|:-:|:-:|:-:|:-:|:-:|:-:|
> |R1\@0.7|58.93±1.71|60.39±0.66|**61.25±0.31**|60.29±0.40|58.20±0.41|59.12±0.89|59.25±0.53|59.06±0.45|
> |mIoU|65.58±1.30|66.18±0.68|**67.16±0.07**|66.44±0.37|64.36±0.54|65.64±0.33|65.56±0.16|65.15±0.43|
> |Acc|84.25±0.57|85.11±0.14|**85.76±0.69**|84.96±0.51|83.50±0.15|84.37±0.28| 84.00±0.51|84.11±0.11|
>
>
> The experimental results indicate that a cluster number of 64 yields optimal performance. This choice provides comprehensive coverage of underlying semantic categories, effectively capturing confounders. In contrast, using too few clusters risks missing critical semantic distinctions, while employing too many introduces computational overhead and noisy representations, negatively impacting model performance.
>
>
> [1] "Deconfounded video moment retrieval with causal intervention." SIGIR 2021.
>
> [2] "DORi: Discovering object relationships for moment localization of a natural language query in a video." WACV 2021.
>
> [3] "Correlation-guided query-dependency calibration for video temporal grounding." arXiv preprint arXiv:2311.08835 (2023).

---

> > ### Comment · Reviewer_JXSa · 2025-08-08
> >
> > Thank you for the detailed response.  Most of my concerns were adequately addressed. After reviewing the feedback from other reviewers, I have decided to maintain my positive score and increase the confidence score to 3.

---

> > > ### Author Response · Authors · 2025-08-08
> > >
> > > We sincerely thank you for your thoughtful comments and valuable time, and we’re glad to hear that our revisions have addressed your concerns.

---

### Public Comment · ~Mingyao_Zhou1 · 2025-11-05
**Video Feature Issues**

I'm delighted to see your interesting work and would like to follow it. Could you please tell me how to obtain the video and text features extracted by CLIP in InternVideo2 used in the paper? Also, I have another question: does the paper report experimental results using the video features (I3D, C3D) and text features GLoVe compared to the baseline?

---

> ### Public Comment · ~Qiyi_Wang1 · 2025-12-09
> **Response to Video Feature Issues**
>
> Thank you for your interest in our work, and we apologize for the delayed response.
>
> Regarding your questions:
>
> 1.  **Obtaining InternVideo2 Features:**
>     For the extraction of video and text features using InternVideo2, please refer to the official guidelines: [InternVideo2 Multi-modality](https://github.com/OpenGVLab/InternVideo/tree/main/InternVideo2/multi_modality#clip-post-pretraining).
>     Additionally, the following issues in their repository provide relevant details that may assist you:
>     * [Issue #129](https://github.com/OpenGVLab/InternVideo/issues/129)
>     * [Issue #268](https://github.com/OpenGVLab/InternVideo/issues/268)
>
> 2.  **Experiments with Other Features (I3D, C3D, GLoVe):**
>     In this paper, we primarily utilized InternVideo2 and CLIP+SlowFast features to validate our framework. We did not report results using legacy features such as I3D, C3D, or GLoVe.
>     However, we have released our full code and checkpoints at [https://github.com/MxLearner/CausalVTG](https://github.com/MxLearner/CausalVTG). The codebase is designed to be extensible, so you are welcome to integrate and test other feature extractors as needed.
>
> Best regards,
> The Authors

---

### Note · Authors · 2025-08-13

We thank all reviewers for their constructive feedback and for acknowledging the novelty and potential impact of CausalVTG. Our revisions and additional experiments have fully addressed all reviewer concerns, with every reviewer maintaining a positive assessment and in some cases increasing their ratings. Below we concisely restate our contributions and summarize the clarifications and additions made during rebuttal.

## Key Contributions:
Our work systematically integrates a structural causal model, modality specific front door adjustment, and counterfactual reasoning into Video Temporal Grounding (VTG). This causal approach directly mitigates stylistic and linguistic confounders and enables reliable query abstention, two long standing and underexplored challenges. Through detailed ablations and sensitivity analyses, we show that each component, CADE, MSTP, and QRM, provides complementary benefits.

## Summary of Rebuttal Discussions:

- **Causal Comparisons:** In response to requests for other causal baselines, we incorporated baselines (e.g., DCM, DoRi, CG-DETR) into our comparisons. Results clearly showed that CausalVTG outperforms all alternatives, affirming the benefits of our front-door and counterfactual design.

- **Mediator Sensitivity:** We conducted a detailed sensitivity analysis on the K-means clustering design used in CADE, showing stable performance across cluster sizes and supporting the causal robustness claim.

- **Component Ablation:** We introduced detailed ablations and joint integration studies across CADE, MSTP, QRM, and QGR, confirming their complementary effects. MSTP contributes to temporal granularity, while CADE plays an irreplaceable role in mitigating spurious co-occurrences and generalizing under style shifts.

- **Generalization & Domain Shift:** To demonstrate CADE’s effectiveness under domain shift, we evaluated on the Charades-CG benchmark, where CADE significantly improved generalization to novel phrases and unseen words.

- **Efficiency & Runtime:** Computational cost analyses showed that CausalVTG remains efficient, Despite the added causal modeling, inference remains competitive, requiring only ~53s on QVHighlights versus 31–69s for baselines.


We believe these additions strengthen both the theoretical soundness and empirical validation of CausalVTG, reinforcing its significance as a robust, generalizable framework for VTG with clear practical implications. We thank the ACs and reviewers again for their engagement.

---

### Decision · Program_Chairs · 2025-09-17

**Decision:**

Accept (poster)

**Comment:**

This paper proposes a novel framework for video temporal grounding that explicitly incorporates causal inference into the model design, consisting of structural causal model, modality-specific front door adjustment, and counterfactual reasoning. Extensive experiments on five widely-used benchmarks demonstrate that it achieves state-of-the-art performance in both localization precision and query relevance detection tasks. This paper received positive ratings from all reviewers; all reviewers agree that the paper tackles an important problem, proposes an interesting approach, and shows significant improvement over previous methods. The rebuttal also addresses most of the reviewer’s concerns, e.g., missing comparisons and analyses. The AC also appreciates the scientific contribution of the novel causal inference framework for video temporal grounding, and thus recommends accepting this work.